**Equilibrium-Approximated Solutions to the Reactive Lauwerier Problem: Thermal Fronts as Controls on Reactive Fronts in Earth Systems** 

#### Roi Roded

Civil, Environmental, and Geo-Engineering, University of Minnesota Twin Cities, Minneapolis, MN, USA

Correspondence to: R. Roded (roi.roded@mail.huji.ac.il)

#### **Abstract**






Rates of subsurface rock alteration by reactive flows are often independent of kinetic rates and governed solely by solute transport. This enables a major simplification that makes models tractable even for complex kinetic systems through the widely applied local equilibrium assumption. Here, this assumption is applied to the Reactive Lauwerier Problem (RLP), which describes non-isothermal fluid injection into a confined aquifer, leading to chemical disequilibrium. Specifically, the thermal changes drive temperature-dependent solubility variations, leading to undersaturation and dissolution or supersaturation precipitation reactions. Using this framework, solutions for reaction rate and porosity evolution are developed and analyzed, yielding a time-dependent criterion for their validity that incorporates time and thermal parameters. A key feature—the coalescence of thermal and reactive fronts—is used to explore their evolution over time in different settings. The applicability of the equilibrium model for important fluid-rock interaction processes is then examined and discussed, including sedimentary reservoir evolution and mineral carbonation in ultramafic rocks. Notably, the approach used here to extend thermal solutions for reactive processes suggests broader applicability. The findings also highlight that thermally driven reactive fronts, particularly near equilibrium, often become stationary after a relatively short period. As a result, their spatial evolution is governed by geological processes operating over much longer timescales.

### 1. Introduction

Natural and anthropogenic systems are often complex, involving intricate interactions between various processes, which makes developing a mechanistic understanding of the system challenging. However, the disparity in timescales between these processes often allows for significant simplification, as one process typically serves as the rate-limiting step that controls the

system's overall evolution. This simplification, in turn, enables the recovery of the system's mechanistic behavior. Such systems range from climate science, where atmospheric and oceanic processes interact and operate at different timescales (Vallis, 2017), to multi-step biochemical processes and enzyme kinetics (Cornish-Bowden, 2013), traffic flow analysis (Lighthill and Whitham, 1955), epidemiology and disease spread (Anderson, 1991), economics (Solow, 1956) and crystal growth (Mullins and Sekerka, 1963).

Similarly, in geothermal systems, thermo-hydro-chemical (THC) processes often involve complex interactions. In particular, geochemical kinetics can be highly intricate, involving multiple species and reactions of varying orders, which are influenced by flow and transport dynamics and thermal variations (Appelo and Postma, 2004; Kolditz et al., 2016; Phillips, 2009). This complexity hinders the understanding of system behaviors and their description using tractable models. However, in many cases, the rate of transport is much slower than the reaction kinetics, effectively controlling the overall reaction rate. These conditions, known as transport-controlled, occur when the transport of reactants or reaction products dictates the reaction rate (Deng et al., 2016; Roded et al., 2020; Steefel and Maher, 2009).



Under transport-controlled conditions, the characteristic timescale of transport,  $t_A$ , is much larger than that of the reaction,  $t_R$ , ( $t_A \gg t_R$ ) and the system is close to chemical equilibrium (i.e., quasi-equilibrium). In such cases, the local equilibrium assumption is often invoked, and the assumption that the reaction rate depends solely on transport allows one to greatly simplify models (Andre and Rajaram, 2005; Lichtner et al., 1996; Molins and Knabner, 2019). The validity of the equilibrium assumption is determined by a large timescale ratio and the Damköhler number, Da, which, assuming a first-order surface reaction, is given by

$$Da = \frac{t_{\rm A}}{t_{\rm R}} = \frac{lA_{\rm S}\lambda}{u_{\rm A}} > 1,\tag{1}$$

where l is a local characteristic length scale,  $u_A$  denotes characteristic Darcy flux [L T<sup>-1</sup>],  $A_s$  is the specific reactive area (L<sup>2</sup> to L<sup>-3</sup> of porous medium) and  $\lambda$  is the kinetic reaction rate coefficient [L T<sup>-1</sup>] (Lichtner et al., 1996; MacQuarrie and Mayer, 2005).

In this study, equilibrium-approximated solutions for geothermal systems are derived. These build upon and extend previous work (Roded et al., 2024b), in which thermally driven reactive transport

solutions were developed within the framework of the *Lauwerier solution* (Lauwerier, 1955). The Lauwerier solution provides an analytical prediction of the thermal field development resulting from the injection of hot (or cold) fluid into a thin, non-reactive, confined layer system (Lauwerier, 1955; Stauffer et al., 2014).

The thermally-driven reactive transport solutions developed by Roded et al. (2024b) integrate temperature-dependent solubility into a reactive flow formulation while incorporating the thermal field based on the Lauwerier solution. Specifically, this setting, referred to as the Reactive Lauwerier Problem (RLP), accounts for thermal variations that drive the system out of geochemical equilibrium, thereby triggering chemical reactions. These disturbances stem from shifts in mineral solubility within groundwater, where thermal fluctuations can induce conditions of either supersaturation or undersaturation. Over time, these thermally-driven reactions lead to changes in rock porosity due to the precipitation, dissolution, or replacement of solid minerals and the associated volumetric changes (Phillips, 2009; Woods, 2015).

Depending on the natural solubility of the minerals in the system, an increase in temperature can lead to either dissolution or precipitation. This occurs because mineral solubilities can either decrease with temperature (*retrograde solubility*) or increase with it (*prograde solubility*; Jamtveit and Yardley, 1996; Phillips, 2009). A notable example includes the prograde solubility of silica, which commonly precipitates in geothermal systems from the cooling of fluids (Pandey et al., 2018; Rawal and Ghassemi, 2014; Taron and Elsworth, 2009). In contrast, carbonate minerals such as calcite and dolomite exhibit an inverse relationship with temperature and retrograde solubility, which is often pronounced and influenced by CO<sub>2</sub> concentration. Depending on the conditions, either rapid dissolution or rapid precipitation can occur in the case of common carbonate minerals (Andre and Rajaram, 2005; Coudrain-Ribstein et al., 1998).

Fluid recharge or injection under constrained physical and chemical conditions, as in RLP settings, is common in both natural and engineered geothermal systems and aquifers (Phillips, 2009; Stauffer et al., 2014). These include aquifer thermal storage, pumping or reinjection of geothermal water, and groundwater storage and recovery applications (Diaz et al., 2016; Fleuchaus et al., 2018; Maliva, 2019), as well as implications for mineral carbonation in mafic or ultramafic rocks (Kelemen et al., 2019; Roded and Dalton, 2024).

In what follows, the settings and equations are first described, which then serve to derive the equilibrium-approximated solutions for the RLP for both radial and planar flows. These solutions are then compared to the reference solutions from Roded et al. (2024b) to validate them and discuss their limitations, along with the derivation of specific criteria for the RLP setup. Next, the evolution of reactive fronts under quasi-equilibrium conditions is analyzed in different settings. Interestingly, it is shown that under certain conditions, thermally driven reactive fronts cease to expand and become essentially stationary after a short timescale, remaining controlled by longer-term tectonic processes. The applicability of the equilibrium model to key processes, including sedimentary aquifer alteration and natural mineral carbonation, is discussed along with an outlook for further theoretical developments.

#### 2. Settings and the Equilibrium Model Equations







This section describes the RLP under the equilibrium assumption and then outlines the specific settings and relevant governing equations. These equations provide the THC equilibrium model (Phillips, 2009; Wood and Hewett, 1982) used to drive the solutions in this work. A comprehensive review of the more general RLP framework and its main assumptions is provided in Roded et al. (2024b) and further revised in Appendix A of this work.

## 2.1. The Equilibrium Reactive Lauwerier Scenario

The Lauwerier problem describes the injection of a hot or cold fluid into a confined aquifer bounded by impermeable bedrock and caprock (Lauwerier, 1955; Stauffer et al., 2014). The fluid flows horizontally along the  $\xi$  coordinate, which can represent either the radial distance (r) in an axisymmetric configuration or the Cartesian coordinate (x) in planar configuration, i.e.,  $\xi = r$  or x. These represent the two primary geometric settings considered in this study. A schematic representation of this system is provided in Fig. 1, with the nomenclature summarized in Appendix E.

Along the flow path downstream from the injection well, heat is transferred between the aquifer and the confining aquiclude layers, which conduct the heat. Within the aquifer, thermal variations influence mineral solubility (i.e., saturation concentration,  $c_s(T)$ ). These solubility changes, in turn, lead to undersaturation and dissolution reactions or, conversely, to supersaturation and precipitation reactions, which modify the aquifer porosity ( $\theta$ ). Porosity changes, whether increases

or decreases, depend on thermal changes (heating or cooling) and the solubility nature of the minerals (prograde or retrograde).

Figure 1. Outline of the Reactive Lauwerier Problem (RLP) under the equilibrium assumption. Hot (or cold) fluid is injected into an aquifer, confined between impermeable bedrock and caprock, at a steady flow rate, Q, and temperature,  $T_{\rm in}$ . The initial temperature is  $T_0$ , and the aquifer thickness is H. Along the flow path, heat from the aquifer conducts through the confining layers. The resulting thermal variations (depicted by color gradients) alter mineral solubility,  $c_s(T)$ , driving chemical reactions that modify aquifer porosity from its initial value,  $\theta_0$ . High Damköhler number conditions and the equilibrium assumption are considered. Under these conditions, the reaction rate,  $\Omega$ , is directly governed by variations in mineral solubility,  $\partial c_s(T)/\partial \xi$ , where  $\xi$  denotes the horizontal coordinate—either the radial coordinate or Cartesian coordinates (i.e.,  $\xi = r$  or x). The vertical coordinate is denoted by z. The reference point for both  $\xi$  and z is at the center of the injection well, which exhibits axial symmetry (as shown in the sketch) or planar symmetry in the Cartesian case (modified after Roded et al. (2024b)).





In this study, the focus is on conditions where reaction kinetics are fast, the Damköhler number is large (Da > 1), and the local equilibrium assumption holds. Under these conditions, the reaction rate,  $\Omega$ , as shown in the next section, can be directly calculated from the thermally driven solubility changes in the system; that is,  $\Omega \propto \partial c_s(T)/\partial \xi$ . Hence, such a solution is independent of the specific reaction kinetics involved.

In terms of geometry and hydrogeological scenarios, the radial setting pertains to injection from a single well or accounts for naturally focused flow of deep-origin fluids through faulted or fractured rock, discharging into a shallower aquifer (Craw, 2000; Micklethwaite and Cox, 2006; Roded et

al., 2013, 2023; Tripp and Vearncombe, 2004). The planar setting describes injection from a row of wells arranged in a straight-line configuration, as initially formulated by Lauwerier (1955).

## 2.2. The Equilibrium-Based Approach





The steady-state, solute advection–reaction equation in the aquifer is:

$$0 = -u\frac{\partial c}{\partial \xi} - \Omega(\xi, t), \tag{2}$$

where  $\xi$  is the horizontal coordinate ( $\xi = r$  or x), u is the Darcy flux, c is the solute concentration and  $\Omega(\xi, t)$  is the reaction rate, which varies in space and time, t (Chaudhuri et al., 2013; Szymczak and Ladd, 2012). In Eq. 2, transient variations are neglected, and the quasi-static approach to reactive flow is applied (see Appendix A and Roded et al. (2024b)).

Defining the solute disequilibrium,  $\Lambda$ , as the difference between the dissolved ion concentration, c, and the temperature-dependent solubility (i.e., saturation concentration),  $c_s(T)$ ,

$$150 \quad \Lambda = c - c_s(T), \tag{3}$$

Eq. 2 can then be rewritten as:

$$0 = -u \left[ \frac{\partial \Lambda}{\partial \xi} + \frac{\partial c_{\rm s}}{\partial \xi} \right] - \Omega(\xi, t). \tag{4}$$

Next, conditions of a high Da number are considered, where reaction rates significantly exceed the rate of advective transport. In this regime, local quasi-equilibrium is maintained along flow paths, and the solute disequilibrium magnitude remains small compared to the overall solubility variation. Specifically,  $\Lambda \ll \Delta c_s$ , where  $\Delta c_s$  denotes the absolute solubility change in the system,  $\Delta c_s = |c_s(T_{in}) - c_s(T_0)|$ , that is, between solubility at the injection temperature,  $T_{in}$ , and at ambient conditions,  $T_0$ .

Under this assumption, the first advective term in Eq. 4 ( $u\partial\Lambda/\partial\xi$ ) becomes negligible compared to the other terms. The governing equation can thus be approximated as (Andre and Rajaram, 2005; Phillips, 2009, p. 237):

$$\Omega(\xi, t) = -u \frac{\partial c_{s}(T)}{\partial \xi}.$$
 (5)

The expression in Eq. 5 provides the THC equilibrium model and demonstrates that, under quasiequilibrium conditions, the solute concentration, c, closely follows the spatially varying solubility determined by the temperature field,  $c_s(T)$ . Notably, it shows that in this regime, the solution for the overall reaction rate,  $\Omega(\xi, t)$ , can be independent of the specific reaction kinetics involved and can be calculated from the solubility gradient.

Lastly, it is noted that the current study focuses on the equilibrium assumption and solves the reduced form given in Eq. 5. This contrasts with the preceding work (Roded et al., 2024b), which focused on solving the full form of Eq. 2 (or Eq. 4) under the assumption of first-order kinetics.

#### 2.3. Initial and Boundary Conditions

The thermal Lauwerier solution incorporates an initial condition of uniform temperature  $T_0$  across the system, along with boundary conditions that specify a constant fluid injection rate at temperature  $T_{in}$  at the injection point ( $\xi = 0$ ). It is assumed that the thickness of the bedrock and caprock, as well as the extent of the aquifer, are infinite.

With respect to the solute transport boundary conditions, the RLP problem is defined by a constant fluid injection rate at temperature  $T_{\rm in}$ , with an initial solute disequilibrium of  $\Lambda = 0$  (i.e., saturated fluid) at the inlet (Roded et al., 2024b). In contrast, the equilibrium-approximated solutions derived from Eq. 5, calculate the reaction rate based on the assumption that it is proportional to the temperature-driven solubility gradient. Consequently, as will be shown in the next section, solute transport boundary conditions are not incorporated. This discrepancy is the focus of the analyses in Section 3.3.

# 3. Results: The Equilibrium Solutions and Their Applicability

#### 3.1. Derivation of the Equilibrium Solutions

#### 3.1.1. Axisymmetric (Radial) Flow

#### 185 Aquifer temperature

The Lauwerier solutions for the temperature distribution in the aquifer (Lauwerier, 1955; Stauffer et al., 2014) serve as the basis for developing the equilibrium-approximated RLP solutions presented here. These solutions are derived by solving the advective heat transport equation in the

aquifer, together with the corresponding conductive heat transfer equation in the confining bedrock and caprock (Eqs. A1–A3 in Appendix A). The solution for axisymmetric settings is given by:

$$T(r,t) = T_0 + \Delta T \operatorname{erfc}[\zeta(r,t)r^2], \tag{6}$$

where erfc is the complementary error function,  $\Delta T = T_{\rm in} - T_0$  is the difference between injection and ambient aquifer temperatures, and  $\zeta$  is defined as:

$$\zeta(r,t) = \frac{\pi \sqrt{K_b C_{p_b}}}{Q C_{p_f} \sqrt{t'}},\tag{7}$$

where Q is the total volumetric flow rate, K is the thermal conductivity, and Cp is the volumetric heat capacity, with the subscripts b and f denoting bulk rock and fluid, respectively. The time variable is defined as  $t' = t - t_{Lg}$ , where  $t_{Lg} = \pi r^2 H C p_b / (C p_f Q)$ , with H denoting the aquifer thickness (see Fig. 1). Assuming uniform flow along the vertical direction (z), the fluid velocity can be calculated from the volumetric flow rate using  $u = Q/(H2\pi r)$ .

The solution of Eq. 6 is valid when t' > 0 (Stauffer et al., 2014), and it is further assumed here that a sufficiently long time has passed such that  $t' \approx t$ . Specifically, the term  $t_{Lg}$  represents a thermal retardation time. It accounts for the delay in the arrival of the thermal front due to advective transport and the thermal energy required to heat the aquifer solid matrix along the flow path (for an analysis of the validity of this assumption, see Roded et al. (2024b)).

Additionally, for simplicity, it is assumed that the heat capacities of both the confining rocks and the aquifer are identical. To account for non-uniform heat capacities, an alternative definition of Eq. 6 can be applied (see Eqs. 3.122 and 3.131, along with the corresponding definitions in Stauffer et al. (2014)).

#### **Thermally Driven Solubility Changes**

The THC equilibrium model in Eq. 5 shows that the reaction rate,  $\Omega(r, t)$ , depends on the thermally driven solubility gradient,  $\partial c_s(T)/\partial r$ . Here, the temperature-dependent solubility is calculated using:

$$c_{s}(T) = c_{s}(T_{0}) + \beta(T - T_{0}), \tag{8}$$

where the parameter  $\beta = \partial c_s/\partial T$ . In Eq. 8, a linear relation between  $c_s$  and T is assumed, with a constant proportionality factor  $\beta$ , which is positive for minerals of prograde solubility and negative for minerals of retrograde solubility (Corson and Pritchard, 2017; Woods, 2015).

In Eq. 5, the derivative of the solubility can be expanded to  $\partial c_s/\partial r = (\partial c_s/\partial T)(\partial T/\partial r)$  and by further substituting the definition  $\beta = \partial c_s/\partial T$ , it can be expressed as,

$$\Omega(r,t) = -u\beta \frac{\partial T}{\partial r}.$$
(9)

The temperature gradient  $\partial T/\partial r$  is calculated by substituting the Lauwerier solution (Eq. 6) and performing differentiation, yielding:

$$\Omega(r,t) = 4u\beta\Delta T \frac{\zeta r}{\sqrt{\pi}} e^{\left(-\zeta^2 r^4\right)}.$$
 (10)

which provides the solution for the reaction rate. The evolution of porosity,  $\theta$ , is described by:

$$\frac{\partial \theta}{\partial t} = -\frac{\Omega(r, t)}{v c_{\text{sol}}},\tag{11}$$

where  $c_{sol}$  is the concentration of soluble solid mineral and v accounts for the stoichiometry of the reaction. Substituting the solution for the reaction rate,  $\Omega$  (Eq. 10), into Eq. 11 and integrating over time yields the solution for the porosity change:

$$\theta(r,t) = \theta_0 - 4u\Delta T \frac{\beta \zeta^2 r^3 t}{v c_{\text{sol}} \sqrt{\pi}} \Gamma\left(-\frac{1}{2}, \zeta^2 r^4\right), \tag{12}$$

where  $\Gamma$  is the incomplete gamma function.

#### 3.1.2. Planar Flow

For the Cartesian case, with injection occurring along a plane, the Lauwerier solution is:

$$T(x,t) = T_0 + \Delta T \operatorname{erfc}[\omega(x,t)x], \tag{13}$$

where  $\omega$  is defined as:

230

$$\omega(x,t) = \frac{\sqrt{K_{\rm b}C_{\rm p_b}}}{HC_{\rm p_f}u\sqrt{t'}},\tag{14}$$

and  $t' = t - t_{Lg}$ , where  $t_{Lg} = xC_{pb}/(C_{pf}u)$ . Similarly to the radial case, it is assumed here that a sufficiently long time has passed such that the condition  $t' \approx t$  applies.

Following steps analogous to those in the radial case, the solutions are derived as:

$$\Omega(x,t) = 2u\Delta T \frac{\beta\omega}{\sqrt{\pi}} e^{(-\omega^2 x^2)},\tag{15}$$

and

$$\theta(x,t) = \theta_0 - 2u\Delta T \frac{\beta \omega^2 xt}{v c_{\text{sol}} \sqrt{\pi}} \Gamma\left(-\frac{1}{2}, \omega^2 x^2\right). \tag{16}$$

## 3.2. Comparison to Reference Solutions (High-Da)

In this section, the results of the equilibrium solutions are compared with the more general solutions to the RLP model, which will henceforth be referred to as the 'reference solutions.' These reference solutions account for far-from-equilibrium conditions and assume surface-controlled reactions and first-order kinetics. The case study considered in the comparison involves a common scenario: dissolution of a fractured carbonate aquifer due to the injection of  $CO_2$ -rich hot water and cooling-driven calcite dissolution. First, the results presented by Roded et al. (2024b) for the reference solutions are briefly summarized to facilitate the comparison with the equilibrium solutions. The reference solutions, along with the case study considered here, are detailed in Roded et al. (2024b). The reference solution equations are also provided in Appendix B, and the parameter values used are listed in Appendix D. These values are identical to those in Roded et al. (2024b), including the radial case flow rate ( $Q = 500 \text{ m}^3 \text{ day}^{-1}$ ).

In Fig. 2, the results of CO<sub>2</sub>-rich hot water injection are shown at successive times since the start of injection. These represent both engineering-relevant conditions (t = 25 yr) and longer geological timescales (t = 10 kyr and 100 kyr), associated with natural processes such as focused deep-origin flow discharging into a shallower aquifer (Craw, 2000; Roded et al., 2023; Tripp and Vearncombe, 2004). The Lauwerier solution and reference solutions are shown by continuous lines (Eqs. 6 and

B2-B3), while the equilibrium solution for the porosity evolution is indicated by circle markers in Fig. 2c (Eq. 12).

Figure 2. Reference solutions for a case study of carbonate aquifer dissolution by cooling hot water, presented for comparison with the equilibrium solution in a radial flow setting. Panels (a)—(c) show temperature (T), solute disequilibrium  $(\Lambda)$ , and porosity  $(\theta)$  plotted as functions of radial position (r) at different times. The continuous lines represent the Lauwerier solution and the reference solutions (Eqs. 6 and B2–B3), while the circles in panel (c) denote the equilibrium solution (Eq. 12). Magnified panels show solute disequilibrium  $(\Lambda)$  and porosity  $(\theta)$  near the inlet region.  $\Lambda$  is scaled by the total solubility variation in the system,  $\Delta c_s$ . The equilibrium solution closely matches the reference solution except near the inlet (see magnified panel and text). Quasi-equilibrium conditions are further supported by the small magnitude of  $\Lambda$ .

During the radial flow within the aquifer, the hot fluid cools by transferring heat into the confining layers, which heat up with time, resulting in the gradual advancement of the thermal front downstream (Fig. 2a). The cooling induces solute disequilibrium ( $\Lambda$ ) associated with undersaturation (note that  $\Lambda$  is negative for undersaturation and positive for supersaturation, see Eq. 3). The magnitude of  $|\Lambda|$  in the aquifer is small compared to the absolute solubility change in the system,  $|\Lambda|/\Delta c_s \ll 1\%$  ( $\Delta c_s = |c_s(T_{in}) - c_s(T_0)|$ ; see Fig. 2b). The small magnitude of

disequilibrium is associated with relatively high CO<sub>2</sub> partial pressure considered (0.03 MPa) and rapid kinetics under these conditions.

Despite its small magnitude, the disequilibrium,  $\Lambda$ , governs the alteration of the aquifer and the evolution of its porosity. Notably, since the water at the inlet is hot and saturated with calcite,  $c = c_s(T_{in})$ , disequilibrium and the reaction rate are zero at the inlet, resulting in no change in porosity (see Fig. 2b and c, along with their magnified views). Disequilibrium (undersaturation) abruptly develops downstream of the injection well, initially forming a small minimum (at  $r \approx 20$  m) before gradually diminishing to zero further downstream.

In accordance with the disequilibrium, the porosity profile sharply increases near the inlet and then gradually decreases downstream (Fig. 2c). Undersaturation and dissolution along the flow path are governed by the interplay of three processes: (I) dissolution, which reduces undersaturation (bringing  $\Lambda$  closer to zero), (II) progressive cooling, which enhances undersaturation, and (III) advection, which transports reaction products (calcium ions) radially outward from the well, sustaining undersaturation. Here, fluid velocity and advection decay with a distance, following a 1/r relationship. Particularly, the thermal changes are also reflected in the time evolution. At an early time (t = 25 yr), disequilibrium and its gradients are relatively high. As the thermal front advances and thermal gradients decrease, the disequilibrium curves flatten.

The equilibrium solution matches the reference solution closely and is violated only near the inlet (r 

**Figure 3.** Reference solutions for a case study of carbonate aquifer dissolution by cooling hot water, presented for comparison with the equilibrium solution in a planar flow setting. Panels (a)–(c) show temperature (T), solute disequilibrium ( $\Lambda$ ), and porosity ( $\theta$ ) as functions of position (x) at different times. The continuous lines represent the thermal Lauwerier solution and the reference solutions (Eqs. 13 and B5–B6), while the circles in panel (c) denote the equilibrium solution (Eq. 16).  $\Lambda$  is scaled by the total solubility variation in the system,  $\Delta c_s$ . Similar to the radial case, the equilibrium solution closely matches the reference solution except near the inlet. This is also supported by the small magnitude of  $\Lambda$ .

For completeness, Fig. 3 presents results for the same case study shown in Fig. 2 under a planar flow setting, with a fluid velocity of  $u = 10^{-6}$  m s<sup>-1</sup>. Similar to the radial case, the equilibrium solution closely matches the reference solution, with deviation occurring only near the inlet (magnification not shown). A key difference from the radial case is that the aquifer is heated over significantly greater distances. This results from the uniform flow velocity and more efficient heat

retention in the planar configuration. In contrast, radial flow involves velocity decay with distance, which increases residence time and enhances conductive heat loss to the surrounding rock.







Additionally, in the radial case, the heat source (e.g., an injection well) acts as a source from which hot fluid spreads outward radially. In contrast, the planar configuration can be conceptualized as injection from a distributed source (e.g., a row of wells) generating a uniform planar front. More precisely, under the perfect thermal mixing assumption, the radial case is treated mathematically as a point source, while the planar case is treated as a line source. Hence, in the radial case, heat conduction is multidirectional, whereas in the planar case, heat is conducted only in vertical directions. These differences influence the temperature profile shape. In the radial case, effective heating near the injection well and later quick decay lead to a sigmoidal (or diffusive front-like) profile, whereas in the linear case there is a decaying profile (cf. Figs. 2 and 3). These differences are further quantified in Section 3.4.

With respect to the results in Figs. 2 and 3, recall that the solutions in Section 3.1 rely on the fundamental assumption of spatial uniformity and symmetry in the reactive flow. However, in practical scenarios, dissolution channels (wormholes) may develop at the reaction front (Chadam et al., 1986; Furui et al., 2022; Roded et al., 2021). These wormholes localize reactive flow, creating heterogeneous flow fields that deviate from the assumed symmetry and uniformity. Consequently, the results in Figs. 2 and 3 represent only an average solution and do not capture local flow variations accurately.

Furthermore, the equilibrium solutions were also found to be applicable to the injection of hot, silica-rich water into a sandstone aquifer, where cooling induces supersaturation, silica precipitation, and porosity reduction, as discussed in Roded et al. (2024b) (not presented). In summary, the results in this section validate the equilibrium solutions against the reference solutions and highlight the inlet advective discrepancy, examined next (Section 3.3). These results also demonstrate their broader applicability across a range of characteristic conditions in natural and applied systems, as further elaborated in the Discussion section.

#### 3.3. Applicability of the RLP Equilibrium Solutions

This section further examines the applicability of equilibrium-approximated solutions, focusing on the inlet advective discrepancy. This is done by considering lower Da, conditions farther from

equilibrium, and changes in the system state over time. Accordingly, a scenario of relatively slow precipitation ( $\beta > 0$ ) is considered, using a kinetic rate coefficient nearly four orders of magnitude lower ( $\lambda = 5 \cdot 10^{-10}$  m/s), while all other conditions remain consistent with Section 3.2. This setup is representative, for example, of carbonate mineral precipitation from water of alkaline composition originating in carbonate or mafic rock aquifers (e.g., basaltic formations). Upon reinjection and subsequent heating, the solubility of carbonate phases decreases, promoting CO<sub>2</sub> mineralization through precipitation reactions (Etiope, 2015; Plummer et al., 1978; Steefel and Lichtner, 1998).

**Figure 4.** Comparison of the reference and equilibrium solutions over time under low Da conditions. (a) Reaction rate,  $\Omega$ , as a function of radial position (r) at different times. The continuous lines represent the reference solution (Eq. B3), and the circles represent the equilibrium solution (Eq. 10), denoted as 'Ref' and 'Equ' in the legend, respectively. (b) The deviation between the solutions, shown using the local error, Err, is visualized as a shaded region. Err is calculated as the radial integral of the difference between the solutions (see text for details).  $\Omega$  and Err are normalized by their maximum values at t = 0.2 kyr, where  $\Omega_{max}$  refers to the reference solution.

Figure 4a presents the results for the reaction rate,  $\Omega$ , for the reference solution (solid lines; Eq. B3) and the equilibrium solution (dashed lines with circle markers; Eq. 10). The slower kinetics and reduced Da result in a significantly larger deviation compared to the case shown in Figs. 2c and 3c. Note that the results in Figs. 2c and 3c, rather, present the porosity evolution, which reflects the time-integrated behavior of  $\Omega$  (see Eq. 11).







Significantly, the peak of the reaction rate curve in Fig. 4a is reached further downstream, rather than occurring immediately near the inlet as observed in Figs. 2 and 3. This shift reflects a much more dominant advective effect but still preserves the same general behavior: advection of saturated fluid from the inlet and the progressive buildup of disequilibrium and elevated  $\Omega$  occur downstream of the injection well. However, in this case, the effect extends over a much greater distance.

Another prominent effect visible in Fig. 4a is the reduction in deviation between the solutions over time. This trend is quantified in Fig. 4b, which shows the weighted local error, Err, calculated from the difference between the two solutions,  $Err = (\Omega_{Ref} - \Omega_{Equ})2\pi r$ , where the subscripts Ref and Equ denote the reference and equilibrium solutions, respectively.

The *Err* shaded regions show a progressive decrease and flattening over time. This reduction in *Err* and the closer approach to equilibrium are attributed to the downstream advancement of the thermal front. As the thermal front advances and extends, the temperature gradients near the inlet become milder. This leads to a decrease in the reaction rate in this region, and the inlet advective discrepancy of the equilibrium model becomes less pronounced (the Supplementary Material presents results for the planar case, which exhibits the same effects).

As noted in the Introduction, the applicability of the equilibrium model is governed by Da, with quasi-equilibrium conditions expected when Da > 1 (Eq. 1). In the specific settings of the RLP, the deviation associated with the equilibrium solutions, which primarily arises from the local inlet effect, evolves over time and is influenced by thermal dynamics. This observation motivated the derivation of a more specific applicability criterion, presented in Appendix C. This analysis is based on a key feature of quasi-equilibrium behavior: the close alignment of the thermal and reactive fronts in the aquifer, which occurs when Da is high (cf. Fig. 2a and b). This behavior is

leveraged to establish a criterion for when the fronts coincide and equilibrium conditions may be assumed. This functional relation, which applies to both planar and radial settings, is given by:

$$1 \gg \frac{2}{\sqrt{\pi t}} \left(\frac{1}{A_{\rm S} \lambda}\right) \left(\frac{\sqrt{K_{\rm b} C_{\rm p_b}}}{H C_{\rm p_f}}\right). \tag{17}$$

In accordance with the results in Fig. 4, the criterion shows that the system approaches equilibrium as time progresses (with a proportionality of  $t^{-1/2}$ ). The second term in the brackets represents the characteristic reaction timescale,  $t_R = 1/A_s\lambda$ , which, in agreement with the high Da condition, indicates that a smaller  $t_R$  leads to faster approach to equilibrium. The final term in the brackets captures the ratio of thermal parameters. When the confining rock's thermal conductivity ( $K_b$ ) and heat capacity ( $C_{pb}$ ) are low, the thermal front advances downstream more rapidly, promoting equilibrium. Similarly, a large product of aquifer thickness and fluid heat capacity ( $HC_{pf}$ ) also facilitates faster thermal front advancement and equilibrium.

Notably, the fluid velocity does not appear in the criterion of Eq. 17. This is attributed to the fact that solute advection enhances disequilibrium (in accordance with the Da criterion), while thermal advection promotes equilibrium by extending and stretching the thermal front. By introducing the fluid velocity, u, and the characteristic length scale, l, into the expression, the criterion in Eq. 17 can be reformulated in terms of two functions:

$$1 \gg f^{-1}g(t),\tag{18}$$

where



420 
$$f = \frac{lA_s\lambda}{u}$$
 and  $g(t) = \frac{2}{\sqrt{\pi t}} \frac{l\sqrt{K_bC_{p_b}}}{uHC_{p_f}}$ . (19)

The first function, f can be referred to as a dynamic Da number and describes the relative effect of reaction versus advective transport. The second function g(t), accounts for the evolution and advancement of the thermal front over time.

The functional criterion in Eqs. 17 and 18, in accordance with the results in Fig. 4, demonstrates that the equilibrium solutions are not applicable as  $t \to 0$  and become less accurate at initial stages.

However, as demonstrated in Fig. 2, the equilibrium-approximated solutions may remain fully valid even at relatively early times. This behavior is observed under common conditions involving fractures carbonate aquifers and silica precipitation, where the validity extends to timescales relevant to engineering applications (e.g., t < 25 yr).

It is recalled here that several inherent assumptions in the Lauwerier solution reduce its accuracy during initial stages (see Appendix A). Moreover, the assumption taken here of negligible thermal retardation time ( $t_{Lg}$ ) and the approximation  $t' \approx t$  employed in the Lauwerier solution affect the accuracy at early times (see Eqs. 6 and 13). This assumption, which is particularly relevant for the radial case, leads to reduced accuracy at early times (e.g., t < 10 years; see Appendix C in Roded et al. (2024b)).

## 3.4. Development of Coalesced Fronts

As mentioned in the previous section, a key feature of quasi-equilibrium behavior is the close alignment of the thermal and reactive fronts in the aquifer, which occurs when the Da is high and reactions dominate over transport. Under these conditions, any disequilibrium induced by thermal changes diminishes rapidly and essentially does not extend downstream of the thermal front, resulting in the coalescence of the fronts. This property is leveraged to infer in a simple manner the spatial distribution and temporal advancement of the coalesced fronts using the thermal Lauwerier solutions.

First, we define the thermal fronts' outer-end positions,  $\zeta_F(t)$ , as the furthest distances of thermal perturbation due to the injection at a given time. The thermal perturbation is quantified by  $\varepsilon = (T(\zeta_F)-T_0)/\Delta T$ , where  $\varepsilon$  is a prescribed small value ( $\varepsilon \ll 1$ ); here,  $\varepsilon = 0.01$ . This threshold uniquely determines the position  $\zeta_F(t)$  at which the temperature perturbation is considered negligible.

Next, rearranging and substituting the definition of  $\varepsilon$  corresponding to the conditions at the fronts' outer-end positions into the Lauwerier solutions (Eqs. 6 and 13) yields:

$$\varepsilon = \operatorname{erfc}(a)$$
, where  $a = \begin{cases} \zeta(t)r_{F}^{2}, & \text{for } \xi = r \\ \omega(t)x_{F}, & \text{for } \xi = x \end{cases}$  (20)

Here, a is a constant determined by  $\varepsilon$ , and for  $\varepsilon = 0.01$ ,  $a \approx 1.8$ . Then, the fronts' outer-end positions can be expressed as:

$$r_{\rm F}(t) = \sqrt{\frac{a}{\zeta(t)}}, \quad \text{and} \quad x_{\rm F}(t) = \frac{a}{\omega(t)}.$$
 (21)

Finally, substituting the definitions of  $\zeta$  and  $\omega$  (Eqs. 7 and 14) into Eq. 21 gives explicit expressions for the advancement of the coalesced fronts under quasi-equilibrium conditions:

$$r_{\rm F}(t) = \sqrt{\frac{aQC_{\rm p_f}}{\pi\sqrt{K_{\rm b}C_{\rm p_b}}}} t^{\frac{1}{4}}, \quad \text{and} \quad x_{\rm F}(t) = \frac{aHC_{\rm p_f}u}{\sqrt{K_{\rm b}C_{\rm p_b}}} t^{\frac{1}{2}}.$$
 (22)

These relations provide a simple way to estimate the spatial positions of the coalesced fronts as a function of time using the thermal solutions alone.

To demonstrate the fronts' advancement, Eqs. 22 are used to plot  $x_F$  and  $r_F$  for three different velocities (u) and flow rates (Q), presented in Fig. 5a and b. This illustrates the decay of the advancement rate over time in both cases: the hot fluid heats the confining rocks as it flows, and the thermal fronts gradually advance downstream. However, due to continuous heat transfer to the confining layers along the flow path, the advancement rate decreases over time and distance.

The key difference between the radial and planar cases, as noted in Section 3.2, is clearly reflected in Eqs. 22 and the results shown in Fig. 5a and b. The planar case exhibits significantly greater heat retention and a higher advancement rate. This is demonstrated by the green dashed lines in Fig. 5a and b, which indicate that half of the final calculated extent,  $1/2x_{\text{Final}}$ , is reached in one quarter of the final time, while in the radial case,  $1/2r_{\text{Final}}$  is approached after one sixteenth of the time. Alternatively, this can be shown by differentiating Eqs. 22 with respect to time, yielding  $\partial r_F/\partial t \propto t^{3/4}$  in the radial case, compared to  $\partial x_F/\partial t \propto t^{1/2}$  in the planar case.

Another case considered here, shown in Fig. 5c and d, is the low-flow-rate limit in radial geometry, where conduction dominates and effectively distributes heat. This is illustrated using two different approaches: (I) the analytical conduction-only solution, representing the limit  $Q \rightarrow 0$  (black lines), and (II) numerical results for low flow rates (Q = 1 and 5 m<sup>3</sup>/day, red and orange curves).

**Figure 5.** Advancement of the coalesced thermal and reactive fronts over time,  $x_F(t)$  and  $r_F(t)$ , for different velocities (u) and flow rates (Q), respectively. Panels (a)—(b) show results for high flow rates, while panels (c)—(d) illustrate the low-flow-rate limit. (a)—(b)  $x_F$  and  $r_F$  are calculated using Eqs. 22. Green dashed lines illustrate the difference between the radial and planar cases: half of the final extents ( $1/2x_{Final}$  and  $1/2r_{Final}$ ) are reached at 1/4 and 1/16 of the final time, respectively. (c)—(d) The low-flow-rate limit refers to the radial case where conduction effectively distributes heat. This is analyzed using solution for conduction-only, representing the limit  $Q \to 0$  (analytical, black lines), and results for low flow rates of Q = 1 and 5 m³/day (numerical, red and orange, respectively). Panel (c) shows  $r_F$  for these cases, while (d) displays the temperature profiles as a function of radial position, r. The black line in (d) represents the conduction-only quasi-steady-state profile, and the colored dashed and continues lines indicate early and later times, respectively, for each flow rate. The close alignment of the lines demonstrates that the thermal field is essentially stationary already at early times. For further details on the calculations, refer to the text.

The analytical solution describes a sphere at constant temperature in an infinite medium, modeling heat conducted from the sphere into the surrounding medium. This time-dependent solution

converges to a quasi-steady-state temperature profile that remains essentially unchanged over time (Stauffer et al., 2014; see details in the SM). The numerical simulations for low flow rates use equations and settings identical to those of the Lauwerier solution but with an important distinction: they do not assume negligible radial conduction. This simplification makes the Lauwerier solution inadequate under conditions of low flow rates and sharp lateral geothermal gradients (see Appendix A). Further details of the numerical calculations are given in Roded et al. (2023).





Figure 5c shows  $r_F$  for the conduction-only case and for Q = 1 and 5 m³/day (other parameter values are consistent with Appendix D). Unlike the high-flow-rate planar and radial cases in Fig. 5a and b,  $r_F$  and the advancement rate essentially level off under these conditions. In particular,  $r_F$  increases with Q but also levels off over time, showing similar behavior to the conduction-only case. This is more clearly shown in Fig. 5d that shows temperature profiles for these cases as a function of radial position,  $r_F$ . It includes the analytical quasi-steady-state temperature profile (conduction-only case) and numerical profiles at low flow rates shown for two consecutive times, with dashed and continues lines indicating early and later times, respectively. The close alignment of the dashed (early time) and continues (later time) lines, and their almost overlap, demonstrate that the temperature profiles change very little after early times. The profiles become nearly stationary over tens to hundreds of years, which is a very brief geological timescale.

The results show effective heat distribution by conduction, with nearly complete cooling occurring within 10–100 m, depending on the flow rate. Overall, both the analytical solution for the limit  $Q \rightarrow 0$  and the numerical solutions at low flow rates demonstrate similar heat transport behavior under these conditions. This scenario of low flow rates is particularly relevant to natural conditions, which often involve low flow rates and can manifest on the surface as low-flow-rate thermal springs (Garven, 1995; Klimchouk et al., 2017; Roded et al., 2013).

These findings have important implications, suggesting that thermally driven reactive fronts can also become nearly stationary, as will be further discussed in the Discussion section. Lastly, it is important to note that the solutions assume an infinite caprock thickness. However, if the thermal front reaches the surface, greater heat exchange between the aquifer and the caprock is expected, which would reduce the thermal front's advancement rate and extent.



#### 4. Discussion and Outlook

#### 4.1. Equilibrium Model Applicability to Hydrothermal Systems

Figure 6 presents an illustrative phase diagram distinguishing between conditions where the THC equilibrium model (Eq. 5) is applicable and those far from equilibrium. The diagram is based on the Damköhler number, which represents the ratio between the characteristic timescales of transport and reaction,  $Da = t_A/t_R$ . The diagonal line marking the transition at Da > 1 ( $Da_{cr}$ ) and hotter colors denote higher Da values and conditions closer to equilibrium. As reactivity ( $1/t_R$ ) increases, the equilibrium model becomes applicable over a wider range of flow velocities, u, or smaller characteristic length scales, l, represented as  $1/t_A = u/l$ . Here, l represents the local characteristic length scale of thermal and solubility variations and accounts for the thermal field effect on reactive transport. Equation 1 assumes first-order kinetics and presents  $Da = l\lambda A_s/u$ , which is useful for quantifying different fluid–rock interactions that can be approximated as following first-order kinetics.

Figure 6. A schematic diagram illustrating the applicability of the THC equilibrium model and the positioning of several notable fluid–rock interaction processes according to their typical reactivity. The diagram is plotted based on the characteristic timescales of reaction and transport that define Da, and shows  $1/t_R$  versus  $1/t_A$  ( $Da = t_A/t_R$ ). The equilibrium model can be assumed when  $Da > Da_{cr}$ , with  $Da_{cr}$  defined as a threshold where  $Da_{cr} > 1$ .  $Da_{cr}$  is represented by the diagonal black line on the diagram, with hot colors indicating high Da values and proximity to equilibrium.

Several notable fluid–rock interaction processes are shown on the diagram, positioned according to their characteristic reactivity. At the top are common carbonates, i.e., limestone and dolomite, which typically exhibit high reaction rates and are highly prone to alteration (with values of  $\lambda$  typically ranging from  $10^{-8}$  to  $10^{-4}$  m/s under engineering applications; Dreybrodt et al., 2005; Peng et al., 2015; Plummer et al., 1978).




Silica precipitation is also prevalent in hydrothermal settings (e.g., quartz vein formation and mineral scaling; Glassley, 2014; Huenges and Ledru, 2011; Oliver and Bons, 2001) and is characterized by relatively high reactivity, with a typical rate constant of  $\lambda = 5 \cdot 10^{-10}$  m/s (Rimstidt and Barnes, 1980). In contrast, while non-crystalline silica (amorphous) precipitates relatively quickly, quartz dissolution is typically slower by several orders of magnitude (Rimstidt and Barnes, 1980). An additional interesting behavior associated with quartz occurs at much higher temperatures (e.g., T > 300 °C), which can prevail near magmatic intrusions. At these high temperatures, quartz exhibits retrograde solubility, which switches to prograde solubility upon cooling (Glassley, 2014; Scott and Driesner, 2018).

Importantly, the specific reactive surface area,  $A_s$ , (L<sup>2</sup> to L<sup>-3</sup> of porous medium) may vary widely across different rock lithologies, and its effect on the applicability of the equilibrium model is comparable to that of kinetics. Specifically,  $A_s$  can vary, e.g., from  $10^{-1}$  m<sup>-1</sup> in fractured rock (Deng and Spycher, 2019; Pacheco and Van der Weijden, 2014) to above  $10^5$  m<sup>-1</sup> for porous medium (Noiriel et al., 2012; Seigneur et al., 2019) and can also evolve during reactive flow (Noiriel, 2015; Seigneur et al., 2019).

The position of these processes on the diagram, supported by calculations in Section 3.2, demonstrates the applicability of the equilibrium model even at relatively high flow rates. This is especially significant, as high flow rates are characteristic of applications such as groundwater storage and recovery, aquifer thermal storage, and geothermal reinjection (Diaz et al., 2016; Fleuchaus et al., 2018; Maliva, 2019).

Additional important settings where thermally driven reactions may play a significant role involve mineral carbonation. In particular, this includes the formation of carbonate veins in ultramafic rocks, such as peridotites, by ascending CO<sub>2</sub>-rich hydrothermal flow (Kelemen et al., 2011; Menzel et al., 2024). The CO<sub>2</sub>-rich fluids first dissolve the rock minerals, primarily olivine. Then, as the

pH rises and cation enrichment occurs, carbonate precipitation, primarily magnesite, takes place further along the upward flow path. The rate-limiting step in the mineral carbonation process is commonly suggested to be the relatively slower kinetics of dissolution compared to precipitation (Hänchen et al., 2006; Kaszuba et al., 2013; Kelemen et al., 2019).







The solubility of olivine is retrograde, as evidenced by the exothermic nature of the reaction (Kaszuba et al., 2013; Prigiobbe et al., 2009). Under such conditions, ascending flow along a decreasing geothermal gradient is expected to promote undersaturation. This continued renewal of undersaturation facilitates the development of an extended, thermally driven dissolution front. Considering the typically low rates of ascending hydrothermal flow (e.g.,  $u < 10^{-7}$  m/s; Garven, 1995), along with characteristic high reaction rates of olivine dissolution at high temperatures (T > 150 °C; Rimstidt, 2015; Rimstidt et al., 2012), it suggests that Da can be large. Consequently, mineral carbonation and vein formation can be controlled by thermally driven solubility changes and described by the THC equilibrium model.

### 4.2. Development of Thermally Driven Reactive Fronts in Earth Systems

The quasi-equilibrium conditions, characterized by the thermal front's control over the reactive front and their coalescence, allowed examination of their evolution in different settings in Section 3.4. A particularly interesting finding is that in radial (or similar) settings, and at relatively low flow rates, a quasi-steady state develops over brief timescales of tens to hundreds of years. Such a cooling process can also produce very steep thermal gradients, as shown in the temperature profile in Fig. 5d, and can cause localized, thermally driven reactive effects. These thermal gradients may be up to two orders of magnitude greater than the typical geothermal gradient resulting from Earth's heat flow (e.g., ~0.025 °C/m; Turcotte and Schubert, 2014).

A relevant example includes hypogenic karst cave formation driven by upwelling hydrothermal flow through a conduit pathway within a fault. This flow discharges and spreads radially in a confined aquifer while cooling rapidly, promoting localized carbonate dissolution around the water inlet (Roded et al., 2023, 2024a). In this case, the results in Fig. 5d suggest that the cave system or alteration front may reach approximately constant final dimensions. These settings may also apply to additional alterations by hypogenic flows and thermal seepages.

Additional relevant settings that can involve coalesced fronts are ascending hydrothermal flow along a decreasing geothermal gradient, leading to cooling and thermally driven reactions. Particularly, as mentioned above (Section 4.1), this may induce olivine dissolution followed by mineral carbonation in veins in ultramafic rocks. Alternatively, quartz vein formation dominantly occurs due to cooling along the flow path (Bons, 2000; Sibson et al., 1975). In these settings, coalesced fronts may become stationary as the hot ascending flow alters the background geothermal gradient, producing a modified steady vertical thermal profile (Person et al., 1996; Roded et al., 2013).

In these cases, where the coalesced, thermally driven reactive front remains stationary over geological timescales, spatial alteration of the front depends on slower tectonic processes. These tectonic timescales are associated with processes such as erosion, subduction, and orogenic activity. A well-known example is the alteration of the geothermal gradient caused by surface erosion or sediment deposition (Haenel et al., 2012; Turcotte and Schubert, 2014). In response to tectonic changes, the slowly varying subsurface thermal field drives the gradual migration of the reactive front.

## 4.3. Theoretical Modeling Outlook







Finally, this study and Roded et al. (2024b) demonstrate the extension of established heat transport solutions to THC-coupled solutions. For future work, the possibility of extending these solutions and approaches in several directions should be investigated. Specifically, it should be examined how the solutions developed can be further extended to address more realistic and complex scenarios. In particular, this includes consideration of more complex kinetic systems involving multiple species and additional or more intricate couplings between variables and parameters.

In such cases, semi-analytical approaches could be especially useful. Due to the quasi-static assumption of reactive flow, the set of equations for reaction rate (Eqs. 10 and 15) or solute disequilibrium (Eqs. B3 and B6) could potentially be implemented in a semi-analytical, coupled, and iterative manner.

Furthermore, the approach taken here and in Roded et al. (2024b) can be adapted to extend additional thermal solutions to significant thermally driven reactive transport scenarios. Notably,

this may be especially practical and feasible under the equilibrium assumption, where thermally driven reactions depend solely on the thermal gradients.

## 5. Summary and Conclusions





In this work, the equilibrium assumption was used to derive thermally driven reactive transport solutions for the RLP (Reactive Lauwerier Problem) in Cartesian and radial coordinates. The solutions were then validated and analyzed against reference solutions and case studies involving thermally driven reactions of carbonates. In particular, the shortcoming of the equilibrium-approximated solutions associated with the advective boundary condition is analyzed. It was found that as the thermal front advances, inlet temperature gradients become milder and the advective discrepancy less pronounced. This also motivated the derivation of a specific functional criterion to describe quasi-equilibrium conditions in the RLP, which incorporates time and thermal parameters and confirms the interpretation.

Following this, a unique feature of quasi-equilibrium conditions—the coalescence of the thermal and reactive fronts—is used to explore their evolution over time. This is examined in both planar and radial settings, and under the low-flow-rate limit where conduction effectively distributes heat. The advancement rate in the radial case decays much more rapidly, and, notably, in the low-flow-rate limit, the front can become essentially stationary within a very short period. Additionally, under these conditions, very sharp temperature gradients are created near the inlet, which can induce localized fluid-rock interactions.

The applicability of the THC equilibrium model for notable fluid–rock interaction processes is then discussed. These include sedimentary reservoir evolution through reactions involving silica and calcite, as well as natural mineral carbonation in ultramafic rocks. These processes are positioned on a phase diagram based on the Damköhler number, illustrating the applicability of the equilibrium model.

Notably, it is suggested that the theoretical approach used here to extend established heat transport solutions to thermally driven reactive transport may be applicable to other important scenarios in Earth systems. Finally, it is emphasized that since thermally driven reactive fronts often become

stationary within a short period, their evolution is governed by geological processes. These processes, such as tectonics or surface erosion and deposition, operate on much longer timescales.

## Appendix A: Underlying Assumptions and Equations of the Equilibrium RLP

This appendix describes the main assumptions of the RLP under the equilibrium assumption. It follows the main presentation from Roded et al. (2024b) and extends it to account for the quasi-equilibrium conditions considered in this study. First, the main assumptions are detailed, followed by a comprehensive overview of the basic conservation equations.

#### A.1. Main Model Assumptions







The thermal Lauwerier (Lauwerier, 1955) solution involves several simplifying assumptions. These include neglecting the initial geothermal gradient and assuming that the basal geothermal heat flux is negligible compared to the heat supplied by the injected fluid. The aquifer is also assumed to be situated at depth, preventing heat from being transferred to the surface; otherwise, there would be greater heat exchange between the aquifer and the caprock. This assumption also depends on the timescale of interest: the thermal front, which rises over time, may not extend to the surface on a short timescale. However, over a longer period, it may transfer heat to the surface, which can be calculated using the characteristic timescale of conduction  $t_C$  ( $t_C = l_C^2/\alpha_b$ , where l accounts for the characteristic length scales of conduction and  $\alpha_b$  is the thermal diffusivity).

In the confining layers, heat is transferred solely through conduction in the vertical direction (z), while neglecting lateral ( $\xi$ ) heat conduction. This assumption restricts the model's applicability to cases with high injected fluid fluxes, where mild lateral temperature gradients evolve. To evaluate the validity of this assumption, a thermal Péclet number is employed, which compares heat advection in the aquifer to lateral heat conduction in the confining layers:  $Pe_T = u_A l/\alpha_b$ , where l is a length scale at which substantial temperature variation occurs (e.g., larger than 2% from the total temperature change,  $\Delta T$ ). A posteriori inspection confirms that  $Pe_T \gg 1$  beyond the initial moments under all conditions considered here. Moreover, after a very short initial phase, the length scale l should exceed the vertical dimension of the aquifer, H, where complete thermal mixing is assumed ( $l \gg H$ ). This assumption may not hold if a thick aquifer (i.e., large H) is considered, and significant vertical temperature gradients are expected to develop.

Additionally, thermal and solute dispersions within the aquifer are neglected, as both thermal ( $Pe_T$ ) and solute ( $Pe_s$ ) Péclet numbers are assumed to be large. Properties of the fluid and solid phases, such as density and thermal conductivity, are assumed to be constant and temperature-independent. Finally, it is assumed that Da > 1, making the equilibrium assumption applicable. As a result, reaction rates are essentially independent of kinetics and reactive surface area, as demonstrated in Section 2.2 of the main text.

#### A.2. The Basic Conservation Equations

#### **Heat Transport:**




Here, the basic conservation equations that underlie the Lauwerier solutions (Eqs. 6 and 13) and the THC equilibrium model (Eq. 5) are presented. More general versions of the conservation equations are provided in Roded et al. (2024b). In what follows, the radial case ( $\xi = r$ ) is considered first, followed by the planar flow case and Cartesian coordinates ( $\xi = x$ ).

Assuming that heat transfer in the radial direction, r, is negligible, the heat equation in the bedrock and caprock confining the aquifer is,

$$\frac{\partial T}{\partial t} = \alpha_b \frac{\partial^2 T}{\partial z^2}, \quad \text{for} \quad \begin{cases} z \le -\frac{H}{2} \\ z \ge \frac{H}{2} \end{cases}$$
 (A1)

where T denotes temperature, t is time, z is the vertical coordinate originating at the center of the injection well and H denotes the aquifer thickness (see Fig. 1). The thermal diffusivity is given by  $\alpha_b = K_b/Cp_b$ , where the subscript b denotes bulk rock, K is the thermal conductivity, and Cp is the volumetric heat capacity (Chen and Reddell, 1983; Stauffer et al., 2014).

Assuming that heat transport in the aquifer is dominated by advection and that perfect mixing prevails in the transverse direction (z), a 'depth-averaged' heat transport equation can be derived for the aquifer domain:

$$C_{\mathrm{p_b}}H\frac{\partial T}{\partial t} = -C_{\mathrm{p_f}}H\frac{1}{r}\frac{\partial (ruT)}{\partial r} - \boldsymbol{n} \cdot \boldsymbol{\Theta}(r,t), \quad \text{for} \quad -\frac{H}{2} \le z \le \frac{H}{2}, \tag{A2}$$

where subscript f denotes fluid and u is the Darcy flux, assumed to be uniform along the z direction, and calculated from the total volumetric flow rate, Q, using  $u(r) = Q/(H2\pi r)$  (Andre and Rajaram,

2005; Lauwerier, 1955). The  $\Theta$  function accounts for the heat exchange between the aquifer and the confining bedrock and caprock, calculated using Fourier's law, assuming continuous temperature at the interfaces:

$$\mathbf{\Theta} = -2K_{\rm b} \frac{\partial T}{\partial z} \Big|_{z = \frac{H}{2}, -\frac{H}{2}}.\tag{A3}$$

The factor of two accounts for both the bedrock and caprock (Stauffer et al., 2014). In Eq. A2, *n* represents a unit vector directed outward from the aquifer and perpendicular to the interface between the aquifer and the bedrock or caprock. This orientation ensures that, e.g., in the case of a warmer aquifer, the upward and downward heat fluxes constitute a heat sink.

#### **Reactive Transport:**


715 The solute advection-reaction equation in the aquifer is:

$$0 = -u\frac{\partial c}{\partial r} - \Omega(r, t), \quad \text{for} \quad -\frac{H}{2} \le z \le \frac{H}{2},\tag{A4}$$

where c is the solute concentration and  $\Omega$  is the reaction rate (Chaudhuri et al., 2013; Szymczak and Ladd, 2012). Note that the transient and dispersivity terms in Eq. A4 are neglected, with the latter being omitted due to the assumption of  $Pe_s \gg 1$ . The justification for neglecting the transient term and invoking the quasi-static approximation in the derivation of Eq. A4, lies in the separation of timescales between the relaxation of solute concentration ( $t_A$ ), heat conduction ( $t_C$ ) in the confining rocks and mineral alteration (for in-depth analysis and discussion see Roded et al. (2024b) and as well, e.g., Bekri et al., 1995; Ladd and Szymczak, 2017; Lichtner, 1991; Roded et al., 2020).

Using the reaction rate, the change in porosity,  $\theta$ , can be calculated as:

$$\frac{\partial \theta}{\partial t} = -\frac{\Omega}{\nu c_{\text{sol}}}, \quad \text{for} \quad -\frac{H}{2} \le z \le \frac{H}{2}.$$
 (A5)

Here,  $c_{sol}$  represents the concentration of soluble solid mineral and v accounts for the stoichiometry of the reaction. For planar flow and Cartesian coordinates, r can be substituted with x in the equations above, and Eq. A2 then takes the following form:

730 
$$C_{p_b}H\frac{\partial T}{\partial t} = -uC_{p_f}H\frac{\partial T}{\partial x} - \boldsymbol{n} \cdot \boldsymbol{\Theta}(x,t), \quad \text{for} \quad -\frac{H}{2} \le z \le \frac{H}{2}.$$
 (A6)

The above set of heat transport equations underlies the development of the thermal Lauwerier solutions presented in Section 3.1 (Eqs. 6 and 13). Section 2.2 of the main text provides the derivation of the equilibrium-approximated form of Eq. A4, which is used to obtain the equilibrium-approximated solutions developed in this study.

## 735 Appendix B: RLP Solutions

#### **B.1. Radial Case**

The solution to the RLP for solute disequilibrium in the radial case is given by,

$$\Lambda = \Delta T \beta e^{\left(\frac{\eta^2}{4\zeta^2} - \eta r^2\right)} \left( \operatorname{erf} \left[ \zeta r^2 - \frac{\eta}{2\zeta} \right] + \operatorname{erf} \left[ \frac{\eta}{2\zeta} \right] \right), \tag{B1}$$

where  $\eta = \pi H A_s \lambda / Q$  and the definition of  $\zeta$  is given in Eq. 7.

A closed-form expression for the temporal and spatial evolution of porosity,  $\theta$ , is given by,

$$\begin{split} \theta(r,t) &= \theta_0 + 4 \frac{\zeta^2 t}{\eta^2} \frac{\lambda A_{\rm s} \Delta T \beta}{\nu c_{\rm sol}} \bigg( -e^{\eta/4 \left(\frac{\eta}{\zeta^2} - 4r^2\right)} \bigg) \bigg( \text{erf} \left[ \zeta r^2 - \frac{\eta}{2\zeta} \right] + \text{erf} \left[ \frac{\eta}{2\zeta} \right] \bigg) + \frac{\eta}{\zeta \sqrt{\pi}} e^{-\eta r^2} \\ &+ \text{erf} [\zeta r^2] (1 - \eta r^2) - \frac{\eta}{\zeta \sqrt{\pi}} e^{-\zeta^2 r^4} + \eta r^2 - 1 \bigg). \end{split} \tag{B2}$$

For efficient computation and preventing integer overflow, an approximate solution of Eq. B1 is developed using the first-order asymptotic expansion of erfc,

745 
$$\Lambda = \frac{\Delta T \beta}{\sqrt{\pi}} e^{(-\eta r^2)} \left( \frac{e^{(\eta r^2 - \zeta^2 r^4)}}{\frac{\eta}{2\zeta} - \zeta r^2} - \frac{2\zeta}{\eta} \right). \tag{B3}$$

#### **B.2.** Planar Case

For the planar case, the corresponding solutions are given by,

$$\Lambda = \Delta T \beta e^{\left(\frac{\sigma^2}{4\omega^2} - \sigma x\right)} \left( \text{erf} \left[ \omega x - \frac{\sigma}{2\omega} \right] + \text{erf} \left[ \frac{\sigma}{2\omega} \right] \right), \tag{B4}$$

and

750 
$$\theta(x,t) = \theta_0 + 4 \frac{\omega^2 t}{\sigma^2} \frac{\lambda A_s \Delta T \beta}{\nu c_{sol}} \left( -e^{\sigma/4 \left( \frac{\sigma}{\omega^2} - 4x \right)} \left( \text{erf} \left[ \omega x - \frac{\sigma}{2\omega} \right] + \text{erf} \left[ \frac{\sigma}{2\omega} \right] \right) + \frac{\sigma}{\omega \sqrt{\pi}} e^{-\sigma x} + \text{erf} \left[ \omega x \right] (1 - \sigma x) - \frac{\sigma}{\omega \sqrt{\pi}} e^{-\omega^2 x^2} + \sigma x - 1 \right).$$
 (B5)

An approximate expression for Eq. B4 is given by

$$\Lambda = \frac{\Delta T \beta}{\sqrt{\pi}} e^{(-\sigma x)} \left( \frac{e^{(\sigma x - \omega^2 x^2)}}{\frac{\sigma}{2\omega} - \omega x} - \frac{2\omega}{\sigma} \right). \tag{B6}$$

Here,  $\sigma = A_s \lambda / u$  and the definition of  $\omega$  is given in Eq. 14.

To prevent integer overflow errors, Eqs. B3 and B6 are used to calculate the undersaturation profiles shown in Figs. 2b and 3b, and the reaction rate profiles in Fig. 4a. These expressions are also used in the iterative numerical solution to obtain the porosity profiles at later times, as shown in Figs. 2c and 3c (t = 100 kyr). Prior validation confirmed the accuracy of the approximate solutions (Eqs. B3 and B6; Roded et al., 2024b).

## **Appendix C: Derivation of the Applicability Criterion**

In this appendix, the derivation of the applicability criterion shown in Section 3.3 is presented. This criterion provides a functional relationship between key parameters, variables, and the system equilibrium state in RLP settings. The derivation of the criterion leverages a key feature of the quasi-equilibrium regime: the coalescence of the thermal and reactive fronts in the aquifer, which occurs when *Da* is high (compare the curves in Fig. 2a and b). In this regime, reactions dominate over transport, and thermally induced disequilibrium dissipates rapidly, essentially not extending downstream of the thermal front.

It is noted that even when the fronts coincide downstream, far-from-equilibrium conditions may still persist upstream. This is observed in the results of Fig. 4, where the equilibrium solution (which aligns with the thermal front) and the reference solution closely match downstream at later times, but diverge upstream. Nonetheless, the derived functional relationships offer useful guidance.

First, the thermal front's outer-end position,  $\xi_F(t)$ , is defined as the furthest distance of thermal perturbation due to the injection at a given time. The thermal perturbation is quantified by  $\varepsilon = (T(\xi_F)-T_0)/\Delta T$ , where  $\varepsilon$  is a prescribed small value ( $\varepsilon \ll 1$ ); here,  $\varepsilon = 0.01$ . Below, we consider the radial case ( $\xi_F = r_F$ ), though applying the same steps to the planar case equations yields the same result.

Rearranging and substituting the definition of  $\varepsilon$  into the Lauwerier solution (Eq. 6) yields:

$$\varepsilon = \operatorname{erfc}(a)$$
, where  $a = \zeta(t)r_F^2$ , (C1)

where a is a constant, and for  $\varepsilon = 0.01$ ,  $a \approx 1.8$ . Then,  $r_F$  can be expressed as,

775

785

$$r_{\rm F} = \sqrt{\frac{a}{\zeta(t)}}.$$
(C2)

Next, an approximate form of the reference solution for disequilibrium is used (Eq. B3 in Appendix B; Roded et al., 2024b). The reasoning for using a far-from-equilibrium-based solution, even though the equilibrium model strictly assumes  $\Lambda = 0$  (cf. Eqs. 4 and 5), is that a small  $\Lambda$  confirms the validity of this approximation. Therefore, solute disequilibrium serves as a metric to quantify the spatial and temporal extent over which the equilibrium assumption is valid.

Assuming quasi-equilibrium at the front's outer-end position,  $r_F$ , and applying the condition  $\varepsilon \ge \Lambda/\Delta c_s$ , where  $\Delta c_s$  denotes the solubility change in the system,  $\Delta c_s = c_s(T_{\rm in}) - c_s(T_0)$ , which here may be positive or negative, Eq. B3 becomes:

$$\varepsilon \ge \frac{\Delta T}{\Delta c_{\rm s}} \frac{\beta}{\sqrt{\pi}} e^{\left(-\eta r_{\rm F}^2\right)} \left( \frac{e^{\left(\eta r_{\rm F}^2 - \zeta^2 r_{\rm F}^4\right)}}{\frac{\eta}{2\zeta} - \zeta r_{\rm F}^2} - \frac{2\zeta}{\eta} \right). \tag{C3}$$

Next, applying a few more steps by substituting the definition from Eq. C2, neglecting early times, and assuming high Da and  $\eta \gg \zeta$ , Eq. C3 can be simplified to:

$$\varepsilon \ge \frac{\Delta T}{\Delta c_s} \frac{\beta}{\sqrt{\pi}} \frac{2\zeta}{\eta}. \tag{C4}$$

Noting that  $\beta = \Delta c_s/\Delta T$  and explicitly substituting the parameters using Eq. 7 and  $\eta = \pi H A_s \lambda/Q$ , 795 Eq. C4 becomes,

$$1 \gg \frac{2}{\sqrt{\pi t}} \left(\frac{1}{A_{\rm s} \lambda}\right) \left(\frac{\sqrt{K_{\rm b} C_{\rm p_b}}}{H C_{\rm p_f}}\right),\tag{C5}$$

where  $A_s$  is the specific reactive area  $[L^{-1}]$  and  $\lambda$  is the kinetic reaction rate coefficient of the first-order reaction  $[L\ T^{-1}]$ . Equation C5 defines the conditions under which the thermal and reactive fronts coincide and provides a functional relationship to the equilibrium state in RLP settings.

## **Appendix D: Parameter Values**

| <b>Table 1.</b> Parameter values used in the simulation in Section 3.2. |                                                                         |
|-------------------------------------------------------------------------|-------------------------------------------------------------------------|
| Aquifer thickness                                                       | H = 4  m                                                                |
| Initial porosity                                                        | $\theta_0 = 0.05$                                                       |
| Total volumetric flow rate <sup>1</sup>                                 | $Q = 500 \text{ m}^3 \text{ day}^{-1}$                                  |
| Fluid velocity                                                          | $u = 10^{-6} \text{ m s}^{-1}$                                          |
| Initial aquifer temperature <sup>2</sup>                                | $T_0 = 20  {}^{\circ}\text{C}$                                          |
| Injection temperature <sup>2</sup>                                      | $T_{\rm in} = 60  ^{\circ}{\rm C}$                                      |
| Fluid volumetric heat capacity <sup>2</sup>                             | $C_{\rm pf} = 4.2 \cdot 10^6 \; {\rm J \; m^{-3} \; {}^{\circ} C^{-1}}$ |
| Rock volumetric heat capacity <sup>2</sup>                              | $C_{\rm pb} = 3.12 \cdot 10^6  \rm J  m^{-3}  {}^{\circ} \rm C^{-1}$    |
| Rock thermal conductivity <sup>2</sup>                                  | $K_{\rm b} = 3 \ {\rm W \ m^{-1} \ ^{\circ} C^{-1}}$                    |
| Calcite rate coefficient <sup>3</sup>                                   | $\lambda = 10^{-6} \text{ m/s}$                                         |
| Fractured carbonates specific reactive surface area <sup>5</sup>        | $A_{\rm s} = 10 \; {\rm m}^{-1}$                                        |
| Calcite mineral concentration <sup>3</sup>                              | $c_{\rm sol} = 2.7 \cdot 10^4  \text{mol m}^{-3}$                       |
| Solubility change parameter calcite <sup>7</sup>                        | $\beta = -0.075 \text{ mol m}^{-3} {}^{\circ}\text{C}^{-1}$             |
| Stoichiometry coefficient <sup>3,4</sup>                                | v = 1                                                                   |

1-Glassley (2014); 2-Huenges and Ledru (2011); 3-Palmer (1991); 4-Rimstidt and Barnes (1980); 5- see Section 4.1; 6-Hussaini and Dvorkin (2021) and Lai et al. (2015); 7-Roded et al. (2023).

800

# **Appendix E: Nomenclature**

| Table 2            | 2. List of Symbols.                                            |                    |                                                                   |
|--------------------|----------------------------------------------------------------|--------------------|-------------------------------------------------------------------|
| Roman              | 1                                                              | $u_{\mathrm{A}}$   | Characteristic fluid velocity, m s <sup>-1</sup>                  |
| a                  | Error function argument                                        | $\boldsymbol{x}$   | Coordinate, m                                                     |
| $A_{\rm s}$        | Specific reactive surface area, m <sup>2</sup> m <sup>-3</sup> | $x_{ m F}$         | Front's outer-end position, planar case, m                        |
| С                  | Solute concentration, mol m <sup>-3</sup>                      | $x_{\text{Final}}$ | Front's outer-end position, planar case, m                        |
| $C_{\rm S}$        | Solubility (saturation concentration), mol m <sup>-3</sup>     | y                  | Coordinate, m                                                     |
| $c_{\mathrm{sol}}$ | Concentration of soluble solid, mol m <sup>-3</sup>            | z                  | Coordinate, m                                                     |
| Cp                 | Volumetric heat capacity, J m <sup>-3</sup> °C <sup>-1</sup>   | Greek              |                                                                   |
| Da                 | Damköhler number                                               | $\alpha$           | Thermal diffusivity, m <sup>2</sup> s <sup>-1</sup>               |
| $Da_{cr}$          | Critical Damköhler number                                      | β                  | Solubility change parameter, mol m <sup>-3</sup> °C <sup>-1</sup> |
| erf                | Error function                                                 | $\Gamma$           | Gamma function                                                    |
| erfc               | Complementary error function                                   | Δ                  | Total difference                                                  |
| Err                | Weighted local error, mol m <sup>-2</sup> s <sup>-1</sup>      | $\varepsilon$      | Number much smaller than one                                      |
| f                  | Equilibrium criterion function                                 | ζ                  | Parameter group, m <sup>-2</sup>                                  |
| g                  | Equilibrium criterion function                                 | η                  | Parameter group, m <sup>-2</sup>                                  |
| Н                  | Aquifer thickness, m                                           | $\theta$           | Porosity                                                          |
| K                  | Thermal conductivity, W m <sup>-1</sup> °C <sup>-1</sup>       | Θ                  | Heat exchange term, W m <sup>-2</sup>                             |
| l                  | Local characteristic length scale, m                           | λ                  | Reaction rate coefficient, m s <sup>-1</sup>                      |
| $l_{\mathrm{C}}$   | Characteristic length scale of conduction, m                   | Λ                  | Solute disequilibrium, mol m <sup>-3</sup>                        |
| n                  | Unit vector                                                    | $\mu$              | Fluid viscosity, Pas                                              |
| p                  | Fluid pressure, Pa                                             | v                  | Stoichiometric coefficient                                        |
| $Pe_{\rm s}$       | Solute Péclet number                                           | ξ                  | Lateral coordinate, $\xi = r$ or $x$ , m                          |
| $Pe_{T}$           | Thermal Péclet number                                          | $\xi_{ m F}$       | Front's outer-end position, $\xi = r_F$ or $x_F$ , m              |
| Q                  | Total volumetric flow rate, m <sup>3</sup> day <sup>-1</sup>   | $\sigma$           | Parameter group, m <sup>-1</sup>                                  |
| r                  | Coordinate, m                                                  | $\omega$           | Parameter group, m <sup>-1</sup>                                  |
| $r_{\mathrm{F}}$   | Front's outer-end position, radial case, m                     | $\Omega$           | Reaction rate, mol m <sup>-3</sup> s <sup>-1</sup>                |
| $r_{\rm Final}$    | Front's front outer-end position, radial case, m               | Subscripts         |                                                                   |
| t                  | Time, s                                                        | b                  | Bulk rock                                                         |
| t'                 | Time parameter, s                                              | Equ                | Equilibrium solution                                              |
| $t_{\mathrm{A}}$   | Characteristic timescale of advection, s                       | f                  | Fluid                                                             |
| $t_{\rm C}$        | Characteristic timescale of conduction, s                      | in                 | Inlet                                                             |
| $t_{ m Lg}$        | Thermal retardation time, s                                    | max                | Max                                                               |
| $t_{ m R}$         | Characteristic timescale of reaction, s                        | Ref                | Reference solution                                                |
| T                  | Temperature, °C                                                | 0                  | Initial average quantity                                          |
| и                  | Fluid velocity, m s <sup>-1</sup>                              |                    |                                                                   |

#### Code & Data availability:

The MATLAB codes and data generated in this study are available to reviewers and upon request from the author.

#### **Competing interests:**

The author declares no competing interests.

#### **Acknowledgments:**

The research was supported as part of the Center on Geo-processes in Mineral Carbon Storage, an Energy Frontier Research Center funded by the U.S. Department of Energy (DOE), Office of Science, Basic Energy Sciences (BES), under Award # DE-SC0023429.

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
