# Peer review of "Equilibrium-Approximated Solutions to the Reactive Lauwerier Problem: Thermal Fronts as Controls on Reactive Fronts in Earth Systems"

_EGUsphere, 2025_

## Referee Comment (RC2)

Please accept my apologies for the delayed review!

Dr. Roded presents an analytical solution-based analysis of reactive, thermal (horizontal, radial and planar) flow resulting from injection into a thin aquifer confined between impermeable rock. This is a follow-up to a previous study (Roded 2024b) and declared to analyze the effect of simplifying the full kinetic treatment with an equilibrium reaction assumption in cases that can be considered transport- rather than reaction rate-controlled.

While the overall results seem to be potentially useful, I have a number of critical remarks on this contribution and I strongly recommend applying major revisions to make this paper more approachable and useful for the community:

1. Simple to correct, but utmost important: the units of the injection rate, Q, immediately appeared wrong. I quickly communicated with the author about this to simplify the review process and, indeed, instead of 500 m3 per *second*, they were taken as 500 m3 per *day* in the reported calculations (i.e., wrong by ca. 4 orders of magnitude). This needs to be corrected in Table 1 (easy) and I strongly recommend to also post a corrigendum to the previous paper where the same mistake occurs.

2. Not so simple and very important: Over large parts of the manuscript the difference to Roded (2024b) is not apparent and the texts are very similar and Figures identical or very similar as well. A revision should make much clearer how the simplifying equilibrium assumption is actually implemented and modifies the previous work. Namely, the derivation of the analytical solutions is essentially identical up to ca. equation 15 and then, without explanation, equations 16 and 17 appear from the same steps of derivation as in the previous paper but are supposed to represent (and look) something different. Here, an explanation of what is the key difference compared to previous work would help a lot (it probably requires very little but I can't really judge based on the information given). It should not be the job of a reader to spend an afternoon or two to re-assess the math and try to find out what is different. Even more importantly is the question to what degree the results are a fortuitous result of the reaction chosen.

3. Following up on that: I sometimes had a hard time understanding in what sense now the "equilibrium" assumption is to be understood. Why is a solution for the reaction rate needed (line 259) and its pre-requisites such as "solute disequilibrium" (eq. 7)? What is the meaning of the latter (also plotted in Fig 2(b)) if equilibrium is supposed to be applicable? Should the arguments in section 3.3 (but see criticism of that section in Point 5 below) come further up-front to make this clearer?

4. The manuscript would very much benefit if the "planar" flow results would also be visualized as the resulting fronts look different from the radial case, i.e., the aquifer can be heated up to the injection temperature for significant distances, displaying a true "front" rather than an outward migrating diffusive-looking profile.
   Then, for both cases it'd be then interesting if a minimal phenomenological description of the underlying physics would be provided why the progressing fronts have the shape that they have. I was a bit surprised that in the radial case it takes very long times to heat even the nearest region around the injection well to injection temperatures. Is that because the radial flow velocities slow

down rapidly with radial distance, enhancing conductive heat loss relative to the planar case?

5. I had a hard time understanding the logic in sections 3.3 and 3.4. First, in my understanding, the main argument on which the analysis rests is not provided but referenced to an appendix of the previous publication (line 410). Why not simply and up-front repeat the key point from there that the thermal font advancement and the reactive front coalescence is simply due to the timescale involved and the nature of the reaction?
Then, there is often unclear writing and wording. For example, what is, in simple words, meant with "thermal front end location"? To me, it sounds like the thermal front comes to a final halt. But then it appears that this is rather meant as the furthest distance of thermal perturbation due to injection at a given time (the outer end of the non-sharp front)? But then, the definition in line 404 and the following manipulations don't seem convincing. First, there is not one position where the criterion is fulfilled but for a given $\epsilon$ the condition is fulfilled for all $r$ greater than a certain value (which I presume is this $r_F$). Second, the choice of $\epsilon$ would be arbitrary and, given the shape of the curve, potentially have a significant impact on the location of $r_F$. I strongly suggest re-writing this whole part in much simpler and more approachable form. Please try to put yourself into the position of the reader who has not done your work and wants to understand what you did.
Another example for strange wording is the "elongation" in section 3.4. Fronts advance and broaden but don't "elongate". So, it seems you mean advancement, which would be a more commonly used term? And certainly not broadening, right? Although in the radial case it is a mixture of both ... Also, as it is written (e.g., lines 443, 447), at least for me, is comes across as if $x_F$ and $r_F$ refer to the thermal and reactive fronts, respectively, while later it appears that these refer to the fronts in the planar and radial case. Please remove these ambiguities.
There are plenty more such examples in the whole text, so a general overhaul towards clarity is highly desirable.

6. Generally, figure captions are too long and often too unclear. Too long: they should not contain explanation that belongs in the text (e.g., Fig 4). Too unclear: e.g., again Fig. 4: why don't you clearly state that panel (a) is for the planar and (b) for the radial case etc.? Or even put a respective label into the respective panels. Even with relatively careful reading it took me way too long to catch what "the low-flow-rate limit assuming conduction only" (panel C) should display. Most other figure captions show similar problems.

7. Very important: what is the relevance of the example calculations for real problems? Injection durations of 200, 10'000 and 100'000 years are unheard of and are very likely to be never implemented (for industrial-scale $CO_2$ storage currently 25 years are typical projections). When extrapolating the magnified part of Fig. 2(c) to shorter times, it seems that one might, for real problems, enter a region where the analytical equilibrium solutions fail to be good approximations. Please discuss this.

8. The "reactions" are simple solubility curves without any actual reaction (e.g., a pH and aqueous speciation in the calcite case is excluded). It'd be desirable to see some reflection how applicable the solutions might be in more complex cases with multiple species and temperature dependencies of equilibrium constants that are less favorable for "coalescence" than in the calcite case. Nature has

examples with multiple reactions fronts progressing at different velocities (contact metamorphic aureoles, skarns in marble layers cross-cut by hydrothermal veins etc.), so these obviously didn't "coalesce" with the thermal front.

To me, this makes the stated applicability to hydrothermal ore deposits questionable (where the author explicitly states: "A particularly intriguing phenomenon, often primarily controlled by the dependence of solubility on temperature, is the zoning of metals and minerals, which is commonly observed at various field scales. In these cases, regular belts of different precipitants form progressively as the distance from the hydrothermal fluid source increases." – so, how does this match the result of the present study with a single such precipitation front aligning with the thermal front being a key outcome?).

This is particularly the case for porphyry copper deposits where the author seems to be not on top of the discussion. Already in 1992, Hemley and Hunt (Econ. Geol., 87, 23-43) specifically eluded on the role of precipitation fronts and their dependence on transport, heat transfer rates etc. Also, the geometry and hydrology of porphyry systems (e.g., Weis et al., Science, 2012) does not compare well with the problem studied here. Interestingly, in porphyry copper deposits, there are transport-limited reaction fronts, seen as alteration halo as these have been studied well (e.g., Cathles and Shannon, EPSL, 2007).

For such reasons, I strongly suggest leaving out much of the rather speculative discussion related to natural examples in its current form. Rather provide some that clearly relate to the problem studied in the sense of Fig. 1 or not mention them.

In spite of this criticism, I found the work quite inspiring. The more it seems important to make it more approachable and clear such that people can benefit from it.

---

## Author Comment (AC1)

**Author Response to Reviewer #1's Comments**
*Manuscript ID: HESS-2025-733*

Below, I provide detailed responses to all comments (quoted verbatim in bold).

- **This is a well-written manuscript that presents simplified, equilibrium-based solutions to the Reactive Lauwerier Problem, which models how thermal changes drive mineral reactions in subsurface aquifers. By assuming reactions are fast compared to fluid transport (i.e., a high Damköhler number), the author derives clear analytical solutions for how porosity and reaction rates evolve. These are shown to agree well with more detailed kinetic models, except very close to the injection point. The paper offers a useful criterion for when the equilibrium assumption is valid and applies the findings to real-world processes like $CO_2$ injection, silica precipitation, and ore formation. The work builds on previous studies and contributes useful insights. I recommend publication after minor clarifications, particularly around what's new compared to the earlier work (Roded et al., 2024b) and how to interpret the model's limitations near injection wells.**

I thank the Reviewer for the thoughtful review and the constructive comments, which are greatly appreciated. The suggestions provided will contribute meaningfully to improving the clarity and structure of the manuscript. Below, I provide detailed responses to each of the Reviewer's comments, along with a description of the planned revisions to the manuscript.

1. **Clarifying the Contribution of the Work:** In line with the Reviewer's comment, and consistent with feedback from Referee #2, the manuscript will be revised to more explicitly differentiate the present study from the earlier work (Roded et al., 2024). To make this distinction clearer, Section 2 ('Settings and Model Equations') will be restructured: most of its current content will be relocated to an appendix, while the main Section 2 will be revised to focus more directly on the derivation and implications specific to the equilibrium model developed in this study. To support this restructuring, a supplementary note—titled *"Supplement to Responses to Referee 1"* and included at the end of this document—has been prepared. This note outlines the derivation of the equilibrium model and will form the core of the revised Section 2.

2.  **Model Limitations Near Injection Well:** I thank the Reviewer for highlighting the important issue of model limitations near the injection well, where the local equilibrium assumption may break down—a point that was also touched on by Referee #2. The manuscript will be revised accordingly. Under high Damköhler number conditions and quasi-equilibrium assumptions, deviations between the equilibrium and kinetic solutions are generally confined to a narrow zone near the injection point (see Fig. 2c). However, in dissolution-dominated cases, these localized deviations may still be significant (see lines 334–341).

    Moreover, at very early times or under conditions farther from equilibrium (i.e., lower Damköhler numbers), the system is more likely to transition into a regime where the assumptions of the analytical equilibrium model no longer hold—particularly near the inlet. This breakdown is illustrated in Fig. 3a and is also captured by the applicability criterion derived in Section 3.2.3 (Eq. 26). This consideration is particularly important in practical geothermal and hydrological contexts, where projection times typically span only several decades. The revised manuscript will explicitly address these limitations of the equilibrium model and clarify its domain of applicability.

**Supplement to Responses to Referee 1: Outline of the Derivation of the Equilibrium Model**

Assuming the reactive Lauwerier problem settings and starting from the stationary, radial solute advection–reaction equation in the aquifer:

$$0 = -u\frac{\partial c}{\partial r} - \Omega(r,t), \qquad (S.1)$$

where $r$ is the radial coordinate, $u$ is the Darcy flux, $c$ is the solute concentration [M L$^{-3}$] and $\Omega(r, t)$ is the reaction rate, which varies in space and time, $t$ (Chaudhuri et al., 2013; Szymczak and Ladd, 2012).

Defining the solute disequilibrium, $\Lambda$, as the difference between the dissolved ion concentration, $c$, and the temperature-dependent solubility (i.e., saturation concentration), $c_s(T)$,

$$\Lambda = c - c_s(T), \qquad (S.2)$$

Equation S.1 can then be rewritten as:

$$0 = -u\left[\frac{\partial \Lambda}{\partial r} + \frac{\partial c_s}{\partial r}\right] - \Omega(r,t). \qquad (S.3)$$

Next, assume high Damköhler number conditions and that the reaction kinetics are fast compared to the advective transport rate. Under these conditions, quasi-equilibrium prevails, and the solute disequilibrium satisfies, $\Lambda \ll \Delta c_s$, where $\Delta c_s$ denotes the absolute solubility change in the system, $\Delta c_s = |c_s(T_{in}) - c_s(T_0)|$, i.e., between $c_s(T_{in})$ at the injection point to $c_s(T_0)$ at ambient conditions. Under these conditions, the first advective term $(-u\partial\Lambda/\partial r)$ becomes negligible compared to the other terms, and Eq. S.3 can be approximated as (Andre and Rajaram, 2005; Phillips, 2009, see p. 237):

$$\Omega(r,t) = u\frac{\partial c_s(T)}{\partial r}. \qquad (S.4)$$

Given an expression for $c_s(T)$ (e.g., Eq. 8 in the main text) and a defined temperature field (e.g., the Lauwerier solution in Eq. 11), a closed-form expression for the reaction rate $\Omega(r, t)$ can be obtained. Notably, this solution for $\Omega(r, t)$ is independent of the specific reaction kinetics involved.

Last, given the solution to Eq. S.4 for the reaction rate, the change in aquifer porosity, $\theta$, can be calculated by solving:

$$\frac{\partial \theta}{\partial t} = -\frac{\Omega(r,t)}{\nu c_{\text{sol}}}, \qquad\qquad (S.5)$$

where $c_{\text{sol}}$ is the concentration of soluble solid mineral and $\nu$ accounts for the stoichiometry of the reaction.

**Remark 1:** The solution for the planar case can be obtained by following the same steps outlined above.

**Remark 2:** The previous work focused on solving the full form of Eq. S.1 (or equivalently, Eq. S.3) without invoking the local equilibrium assumption. In contrast, the current approach solves the reduced form given in Eq. S.4.

**References**

Andre, B. J. and Rajaram, H.: Dissolution of limestone fractures by cooling waters: Early development of hypogene karst systems, Water Resour. Res., 41, 2005.

Chaudhuri, A., Rajaram, H., and Viswanathan, H.: Early-stage hypogene karstification in a mountain hydrologic system: A coupled thermohydrochemical model incorporating buoyant convection, Water Resour. Res., 49, 5880–5899, 2013.

Phillips, O. M.: Geological fluid dynamics: sub-surface flow and reactions, Cambridge University Press, 2009.

Roded, R., Aharonov, E., Szymczak, P., Veveakis, M., Lazar, B., and Dalton, L. E.: Solutions and case studies for thermally driven reactive transport and porosity evolution in geothermal systems (reactive Lauwerier problem), Hydrol. Earth Syst. Sci., 28, 4559–4576, 2024.

Szymczak, P. and Ladd, A. J. C.: Reactive-infiltration instabilities in rocks. Fracture dissolution, J. Fluid Mech., 702, 239–264, 2012.

---

## Author Comment (AC2)

**Author Response to Reviewer #2's Comments**
*Manuscript ID: hess-2025-733*

Below, I provide detailed responses to all comments (quoted verbatim in bold).

> **Dr. Roded presents an analytical solution-based analysis of reactive, thermal (horizontal, radial and planar) flow resulting from injection into a thin aquifer confined between impermeable rock. This is a follow-up to a previous study (Roded 2024b) and declared to analyze the effect of simplifying the full kinetic treatment with an equilibrium reaction assumption in cases that can be considered transport- rather than reaction rate-controlled.**
>
> **While the overall results seem to be potentially useful, I have a number of critical remarks on this contribution and I strongly recommend applying major revisions to make this paper more approachable and useful for the community:**

I am grateful to the Reviewer for his careful review and for the important and constructive comments. The detailed conceptual and technical feedback is highly appreciated and will significantly contribute to improving the manuscript and enhancing its clarity. Below are the responses to the comments and the corresponding plan for the intended revisions.

**Reviewer's Comment (1):**

> **Simple to correct, but utmost important: the units of the injection rate, Q, immediately appeared wrong. I quickly communicated with the author about this to simplify the review process and, indeed, instead of 500 m3 per second, they were taken as 500 m3 per day in the reported calculations (i.e., wrong by ca. 4 orders of magnitude). This needs to be corrected in Table 1 (easy) and I strongly recommend to also post a corrigendum to the previous paper where the same mistake occurs.**

I thank the Reviewer for catching this important typographic error in the table (the in-text value in the previous work and in Fig. 4 of the current manuscript correctly indicate the intended values).

***Summary of proposed revisions:***

- Table 1 will be corrected in the revised manuscript.

- A corrigendum will be submitted to HESS for the previous publication.

**Reviewer's Comment (2):**

> **Not so simple and very important: Over large parts of the manuscript the difference to Roded (2024b) is not apparent and the texts are very similar and Figures identical or very similar as well. A revision should make much clearer how the simplifying equilibrium assumption is actually implemented and modifies the previous work. Namely, the derivation of the analytical solutions is essentially identical up to ca. equation 15 and then, without explanation, equations 16 and 17 appear from the same steps of derivation as in the previous paper but are supposed to represent (and look) something different. Here, an explanation of what is the key difference compared to previous work would help a lot (it probably requires very little but I can't really judge based on the information given). It should not be the job of a reader to spend an afternoon or two to re-assess the math and try to find out what is different. Even more importantly is the question to what degree the results are a fortuitous result of the reaction chosen.**

I fully agree with the Reviewer that the distinction from the prior work should be made clearer. To both highlight the contribution of the current study and avoid repetition, I propose relocating most of Section 2 ('Settings and Model Equations') to an appendix, while revising the main Section 2 to focus more directly on the elements relevant to the present work. This restructuring will reduce redundancy in the main text while keeping the full model description readily accessible to readers.

Specifically, the *"Supplement to Responses to Referee 2"*, which outlines the derivation of the equilibrium model, appears at the end of this document. This short derivation will comprise the main focus of the revised Section 2. I believe this adjustment, based on the attached supplement, helps clarify how the simplifying equilibrium assumption is implemented and how it differs from the framework established in the previous work.

Furthermore, with respect to the similarity to the previous study, it is noted that—had it not been for the time gap—this study and Roded et al. (2024) could have been published together as companion papers. Since this was not the case, and in order to provide the necessary background and ensure the manuscript stands on its own, some repetition of the model settings and equations has been included, though direct text duplication was avoided. Additionally, Section 3.2.1 ('Comparison to the Reference Solution') provides brief background information to facilitate the comparison between the results of the

newly developed solutions and those of the previous work (the 'reference solutions'). I am also open to making changes to Fig. 1; however, this figure simply outlines the Reactive Lauwerier Problem (RLP) and cites the source of the previous open-access study.

**Summary of proposed revisions:**

- Relocate substantial portions of Section 2 to an Appendix, while revising the main body of Section 2 to focus on the key equations and aspects directly relevant to the current study.

- Revise the analytical derivations for clarity and show the logic of the approach, and how it specifically differs from the derivations presented in the previous paper.

**Reviewer's Comment (3):**

**Following up on that: I sometimes had a hard time understanding in what sense now the "equilibrium" assumption is to be understood. Why is a solution for the reaction rate needed (line 259) and its pre-requisites such as "solute disequilibrium" (eq. 7)? What is the meaning of the latter (also plotted in Fig. 2(b)) if equilibrium is supposed to be applicable? Should the arguments in Section 3.3 (but see criticism of that section in Point 5 below) come further up-front to make this clearer?**

I thank the Reviewer for pointing out that these issues were not made clear. I believe that the proposed revision of Section 2—outlined at the end of this document under the heading *"Supplement to Responses to Referee 2"*—will help clarify these points.

Moreover, the manuscript will be revised to explain:

- Under the equilibrium assumption, the solution is derived for the reaction rate, $\Omega(\xi, t)$ where $\xi = r$ or $x$, rather than for the solute disequilibrium (or concentration). This is because the equilibrium model renders the inclusion of detailed kinetics unnecessary.

- For high Damköhler conditions and fast kinetics, quasi-equilibrium prevails, and $\Lambda \ll \Delta c_s$, where $\Delta c_s$ denotes the absolute solubility change in the system ($\Delta c_s = |c_s(T_{in}) - c_s(T_0)|$ )|, that is, between $c_s(T_{in})$ at the injection point to $c_s(T_0)$ at ambient conditions). Thus, solute disequilibrium is used to quantify the extent to which equilibrium holds across space and time (see Figs. 2b and 3a). Specifically, even within an equilibrium-assumption framework, it is useful to evaluate the

disequilibrium, $\Lambda(\xi, t)$, to assess model applicability—particularly near the injection well, where deviations from equilibrium may emerge. The disequilibrium-based solution (Eq. B.3 of Appendix B) also serves in developing the applicability criterion for the equilibrium approximation, as presented in Section 3.3.

Lastly, following the proposed revisions intended to improve clarity, it may be preferable to retain the current order of presentation—beginning with the results in Figs. 2 and 3, followed by the derivation of the applicability criterion in Section 3.3—as this sequence appears to offer a clear and logical progression.

**Reviewer's Comment (4):**

**The manuscript would very much benefit if the "planar" flow results would also be visualized, as the resulting fronts look different from the radial case, i.e., the aquifer can be heated up to the injection temperature for significant distances, displaying a true "front" rather than an outward migrating diffusive-looking profile. Then, for both cases, it'd be interesting if a minimal phenomenological description of the underlying physics would be provided—why the progressing fronts have the shape that they have. I was a bit surprised that in the radial case it takes very long times to heat even the nearest region around the injection well to injection temperatures. Is that because the radial flow velocities slow down rapidly with radial distance, enhancing conductive heat loss relative to the planar case?**

I agree that including the planar case in the results could be valuable. While the primary focus has been on the radial case, due to its greater relevance to geothermal systems and natural settings, the planar configuration is also of interest.

The Reviewer is correct in noting that, in the planar case, the aquifer can be heated over significantly greater distances. Regarding the difference in thermal front characteristics, the velocity decay in radial geometry indeed enhances conductive heat loss relative to the planar case. Specifically, in radial flow, the fluid velocity decreases with distance from the injection point, which leads to longer residence times and increased opportunity for heat to dissipate into the surrounding rock.

An additional contributing factor is the difference in heat conduction geometry. In the radial case, the heat source is effectively a vertical line (e.g., an injection well), with hot fluid spreading outward in all directions. In contrast, the planar configuration can be conceptualized as injection from a distributed source—such as a row of wells—generating

a nearly uniform planar front. This planar setup facilitates more efficient heat retention and results in a front that advances farther downstream. Of course, in both cases, the thermal front behavior is also influenced by flow rates and the volume of injected hot fluid.

While this is not the central focus of the current work and may have been addressed in previous studies (e.g., Chen & Reddell, 1983; Stauffer et al., 2014; Ziagos & Blackwell, 1986), it is discussed in Section 3.4 ('Development of Coalesced Fronts').

In that section, the the thermal front outer-end (i.e., the furthest distance of thermal perturbation) is denoted as $\xi_F = r_F$ or $x_F$ for the radial and planar cases, respectively. Specifically, it is identified as the downstream point where the temperature slightly deviates from the ambient, using the relation $\varepsilon = (T(\xi_F)-T_0)/\Delta T$. Here, $\varepsilon$ is a prescribed small value ($\varepsilon \ll 1$) and $\Delta T = T_{in} - T_0$ is the temperature difference between injected and ambient fluids. Equations 29 and Fig. 4 present the evolution of $r_F$ and $x_F$ and illustrate the difference between the two cases.

Specifically, the *advancement rate* of the thermal front outer-end (i.e., $\partial \xi_F/\partial t$) decays significantly faster in the radial case: the derivative of the front location scales as $t^{-3/4}$, whereas in the planar case it follows $t^{-1/2}$. This analytical result highlights the fundamental physical difference in front propagation between the two geometries.

**Summary of proposed revisions:**

- Add results for the planar case to the supplementary material, similar to those presented in Fig. 2.

- Update Fig. 3 to include the planar case, illustrating the system's state evolution over time and the applicability of the equilibrium solution (by duplicating the current insets for the planar configuration).

*Note: These revisions may be further adjusted based on the Reviewer's feedback.*

**Reviewer's Comment (5):**

**I had a hard time understanding the logic in sections 3.3 and 3.4. First, in my understanding, the main argument on which the analysis rests is not provided but referenced to an appendix of the previous publication (line 410). Why not simply and up-front repeat the key point from there that the thermal font**

**advancement and the reactive front coalescence is simply due to the timescale involved and the nature of the reaction?**

**Then, there is often unclear writing and wording. For example, what is, in simple words, meant with "thermal front end location"? To me, it sounds like the thermal front comes to a final halt. But then it appears that this is rather meant as the furthest distance of thermal perturbation due to injection at a given time (the outer end of the non-sharp front)? But then, the definition in line 404 and the following manipulations don't seem convincing. First, there is not one position where the criterion is fulfilled but for a given $\epsilon$ the condition is fulfilled for all $r$ greater than a certain value (which I presume is this $rF$). Second, the choice of $\epsilon$ would be arbitrary and, given the shape of the curve, potentially have a significant impact on the location of $rF$. I strongly suggest re-writing this whole part in much simpler and more approachable form. Please try to put yourself into the position of the reader who has not done your work and wants to understand what you did.**

**Another example for strange wording is the "elongation" in section 3.4. Fronts advance and broaden but don't "elongate". So, it seems you mean advancement, which would be a more commonly used term? And certainly not broadening, right? Although in the radial case it is a mixture of both ... Also, as it is written (e.g., lines 443, 447), at least for me, is comes across as if $xF$ and $rF$ refer to the thermal and reactive fronts, respectively, while later it appears that these refer to the fronts in the planar and radial case. Please remove these ambiguities. There are plenty more such examples in the whole text, so a general overhaul towards clarity is highly desirable.**

I thank the reviewer for raising these critical points, and I will carefully revise Sections 3.3 and 3.4 to improve clarity and make the content more accessible.

Regarding the main argument of the analysis: the first paragraph of Section 3.3 will be revised to more clearly and explicitly present the central reasoning. Specifically, it will emphasize that the coalescence of the reactive front and the thermal front is governed by the timescales involved and high Damköhler number conditions. The coalesced fronts then migrate in response to the thermal field evolution. With respect to the reference to the appendix of the previous publication in line 410, it is intended only to cite the specific equation used in the analysis, not to support the main conceptual argument.

Concerning the definition of $\varepsilon$ in line 404 (i.e., $\varepsilon = (T(r_F) - T_0)/\Delta T$, where $\varepsilon \ll 1$), I will revise the text to explicitly state that $\varepsilon$ refers to a specific, prescribed small value (here taken as $\varepsilon = 0.01$) that is used consistently in the analysis and substituted into Eqs. 11 and 18. For a given prescribed value of $\varepsilon$ (and hence of $a$), the definition of $r_F$ in Eq. 23 becomes unique.

While the choice of $\varepsilon$—restricted to values where $\varepsilon \ll 1$—is somewhat arbitrary, the resulting expressions (Eqs. 26 and 27) remain valid and unaffected. Furthermore, the results shown in Fig. 4a and 4b, which are based on Eqs. 29, are not expected to change appreciably depending on the exact choice of $\varepsilon$. This section will be rewritten in a clearer and more accessible form to ensure these points are clearly communicated.

I will also revise the terminology and replace the word "elongation" throughout Section 3.4 with "advancement," which more accurately reflects the intended meaning. Additionally, the ambiguity surrounding the definition of the front outer-end position variables—$x_F$ and $r_F$, in the planar and radial cases, respectively—will be addressed by clearly and consistently defining these terms upon their first mention.

**Reviewer's Comment (6):**

> **Generally, figure captions are too long and often too unclear. Too long: they should not contain explanation that belongs in the text (e.g., Fig 4). Too unclear: e.g., again Fig. 4: why don't you clearly state that panel (a) is for the planar and (b) for the radial case etc.? Or even put a respective label into the respective panels. Even with relatively careful reading it took me way too long to catch what "the low-flow-rate limit assuming conduction only" (panel C) should display. Most other figure captions show similar problems.**

Figure captions throughout the manuscript will be shortened and revised to enhance clarity. Specifically, I fully agree with the Reviewer that panel (c) of Fig. 4, which refers to the low-flow-rate limit, along with its associated text, is unclear and will be carefully revised to improve clarity.

**Reviewer's Comment (7):**

> **Very important: what is the relevance of the example calculations for real problems? Injection durations of 200, 10'000 and 100'000 years are unheard of and are very likely to be never implemented (for industrial-scale $CO_2$ storage currently 25 years are typical projections); when extrapolating the magnified**

**part of Fig. 2(c) to shorter times, it seems that one might, for real problems, enter a region where the analytical equilibrium solutions fail to be good approximations. Please discuss this.**

I thank the Reviewer for these important comments. The manuscript will be revised to emphasize that the modeled scenario is not limited to engineered injection settings but is also relevant to natural systems (as noted in lines 132–133 of Section 2.1.1), where flow and reactions can occur over geological timescales. Additionally, the manuscript will be updated to include simulation results for a 25-year duration in Fig. 2 (replacing the current 200-year curve).

As the Reviewer correctly notes, at earlier times, the system is more likely to enter a regime in which the assumptions of the analytical equilibrium model break down—particularly near the injection point—as also indicated by the applicability criterion derived in Eq. 26. The manuscript will be updated to explicitly discuss these limitations of the equilibrium solutions in the context of applied geothermal and hydrological systems.

**Reviewer's Comment (8):**

**The "reactions" are simple solubility curves without any actual reaction (e.g., a pH and aqueous speciation in the calcite case is excluded). It'd be desirable to see some reflection how applicable the solutions might be in more complex cases with multiple species and temperature dependencies of equilibrium constants that are less favorable for "coalescence" than in the calcite case. Nature has examples with multiple reactions fronts progressing at different velocities (contact metamorphic aureoles, skarns in marble layers cross-cut by hydrothermal veins etc.), so these obviously didn't "coalesce" with the thermal front. To me, this makes the stated applicability to hydrothermal ore deposits questionable (where the author explicitly states: "A particularly intriguing phenomenon, often primarily controlled by the dependence of solubility on temperature, is the zoning of metals and minerals, which is commonly observed at various field scales. In these cases, regular belts of different precipitants form progressively as the distance from the hydrothermal fluid source increases." – so, how does this match the result of the present study with a single such precipitation front aligning with the thermal front being a key outcome?). This is particularly the case for porphyry copper deposits where the author seems to be not on top of the discussion. Already in 1992, Hemley and Hunt (Econ. Geol.,**

**87, 23–43) specifically eluded on the role of precipitation fronts and their dependence on transport, heat transfer rates etc. Also, the geometry and hydrology of porphyry systems (e.g., Weis et al., Science, 2012) does not compare well with the problem studied here. Interestingly, in porphyry copper deposits, there are transport-limited reaction fronts, seen as alteration halo as these have been studied well (e.g., Cathles and Shannon, EPSL, 2007). For such reasons, I strongly suggest leaving out much of the rather speculative discussion related to natural examples in its current form. Rather provide some that clearly relate to the problem studied in the sense of Fig. 1 or not mention them.**

Due to the simplifications required to derive closed-form expressions, analytical solutions typically apply directly only to a narrow set of real-world cases—if at all. However, they are valuable for establishing fundamental conceptual understanding, identifying key functional relationships, and also serving as important benchmark cases for validating numerical models (which are more useful for exploring real-world scenarios).

I thank the Reviewer for noting in detail that the scenario considered in this study is, however, too simplified to support extrapolated conclusions regarding ore deposit formation.

With respect to the applicability of the solutions to more complex cases—such as those involving multiple species and temperature-dependent equilibrium constants less favorable to coalescence—this applicability is limited. I believe that such systems cannot be adequately captured using the current equilibrium-approximated solutions. However, I am considering whether the set of equations for solute disequilibrium, such as B.1 or B.3 (Appendix B), can be implemented in a semi-analytical, coupled and iterative manner. This approach could potentially represent solutions for a multi-species system operating far from equilibrium, as these equations account for first-order reaction kinetics.

Similarly, under quasi-equilibrium conditions, it may be feasible to employ the equations presented in Eq. 16 or 20 in a semi-analytical, coupled form to model multi-species behavior. Nonetheless, the validity of such an approach remain subject to further evaluation.

Furthermore, the development of more advanced solution frameworks that incorporate additional kinetic and thermodynamic couplings remains a direction for future work. This outlook, along with its associated challenges, will be briefly discussed in the revised manuscript.

***Summary of proposed revisions:***

- Excluding the discussion related to ore-formation.

- Add a discussion outlining possible pathways for extending the solutions presented in this work and the previous study to more complex kinetic systems and to account for additional or more intricate inter-couplings.

**Final Reviewer's Comment:**

**In spite of this criticism, I found the work quite inspiring. The more it seems important to make it more approachable and clear such that people can benefit from it.**

I once again thank the Reviewer for the careful review, insightful and important comments, and positive attitude. This will certainly help improve the manuscript and enhance its clarity and accessibility for a broader readership.

**Supplement to Responses to Referee 2: Outline of the Derivation of the Equilibrium Model**

Assuming the reactive Lauwerier problem settings and starting from the stationary, radial solute advection–reaction equation in the aquifer:

$$0 = -u\frac{\partial c}{\partial r} - \Omega(r,t), \tag{S.1}$$

where $r$ is the radial coordinate, $u$ is the Darcy flux, $c$ is the solute concentration [M L$^{-3}$] and $\Omega(r,t)$ is the reaction rate, which varies in space and time, $t$ (Chaudhuri et al., 2013; Szymczak and Ladd, 2012).

Defining the solute disequilibrium, $\Lambda$, as the difference between the dissolved ion concentration, $c$, and the temperature-dependent solubility (i.e., saturation concentration), $c_s(T)$,

$$\Lambda = c - c_s(T), \tag{S.2}$$

Equation S.1 can then be rewritten as:

$$0 = -u\left[\frac{\partial \Lambda}{\partial r} + \frac{\partial c_s}{\partial r}\right] - \Omega(r,t). \tag{S.3}$$

Next, assume high Damköhler number conditions and that the reaction kinetics are fast compared to the advective transport rate. Under these conditions, quasi-equilibrium prevails, and the solute disequilibrium satisfies, $\Lambda \ll \Delta c_s$, where $\Delta c_s$ denotes the absolute solubility change in the system, $\Delta c_s = |c_s(T_{in}) - c_s(T_0)|$, i.e., between $c_s(T_{in})$ at the injection point to $c_s(T_0)$ at ambient conditions. Under these conditions, the first advective term $(-u\partial\Lambda/\partial r)$ becomes negligible compared to the other terms, and Eq. S.3 can be approximated as (Andre and Rajaram, 2005; Phillips, 2009, see p. 237):

$$\Omega(r,t) = u\frac{\partial c_s(T)}{\partial r}. \tag{S.4}$$

Given an expression for $c_s(T)$ (e.g., Eq. 8 in the main text) and a defined temperature field (e.g., the Lauwerier solution in Eq. 11), a closed-form expression for the reaction rate $\Omega(r,t)$ can be obtained. Notably, this solution for $\Omega(r,t)$ is independent of the specific reaction kinetics involved.

Last, given the solution to Eq. S.4 for the reaction rate, the change in aquifer porosity, $\theta$, can be calculated by solving:

$$\frac{\partial \theta}{\partial t} = -\frac{\Omega(r,t)}{v c_{\text{sol}}}, \qquad\qquad (S.5)$$

where $c_{\text{sol}}$ is the concentration of soluble solid mineral and $v$ accounts for the stoichiometry of the reaction.

**Remark 1:** The previous work focused on solving the full form of Eq. S.1 (or equivalently, Eq. S.3) without invoking the local equilibrium assumption. In contrast, the current approach solves the reduced form given in Eq. S.4.

**Remark 2:** The solution for the planar case can be obtained by following the same steps outlined above.

**References**

Andre, B. J. and Rajaram, H.: Dissolution of limestone fractures by cooling waters: Early development of hypogene karst systems, Water Resour. Res., 41, 2005.

Chaudhuri, A., Rajaram, H., and Viswanathan, H.: Early-stage hypogene karstification in a mountain hydrologic system: A coupled thermohydrochemical model incorporating buoyant convection, Water Resour. Res., 49, 5880–5899, 2013.

Chen, C. and Reddell, D. L.: Temperature distribution around a well during thermal injection and a graphical technique for evaluating aquifer thermal properties, Water Resour. Res., 19, 351–363, 1983.

Phillips, O. M.: Geological fluid dynamics: sub-surface flow and reactions, Cambridge University Press, 2009.

Stauffer, F., Bayer, P., Blum, P., Molina-Giraldo, N., and Kinzelbach, W.: Thermal use of shallow groundwater, 2014.

Szymczak, P. and Ladd, A. J. C.: Reactive-infiltration instabilities in rocks. Fracture dissolution, J. Fluid Mech., 702, 239–264, 2012.

Ziagos, J. P. and Blackwell, D. D.: A model for the transient temperature effects of horizontal fluid flow in geothermal systems, J. Volcanol. Geotherm. Res., 27, 371–397, 1986.

---

## Author Response (AR1)

**Manuscript ID: HESS-2025-733**

**Author Response to Reviewer #1's Comments**

This Author Response includes detailed responses to the Reviewer's comments, quoted verbatim in **bold**, along with the corresponding revisions made to the manuscript.

• This is a well-written manuscript that presents simplified, equilibrium-based solutions to the Reactive Lauwerier Problem, which models how thermal changes drive mineral reactions in subsurface aquifers. By assuming reactions are fast compared to fluid transport (i.e., a high Damköhler number), the author derives clear analytical solutions for how porosity and reaction rates evolve. These are shown to agree well with more detailed kinetic models, except very close to the injection point. The paper offers a useful criterion for when the equilibrium assumption is valid and applies the findings to real-world processes like CO₂ injection, silica precipitation, and ore formation. The work builds on previous studies and contributes useful insights. I recommend publication after minor clarifications, particularly around what's new compared to the earlier work (Roded et al., 2024b) and how to interpret the model's limitations near injection wells.

I thank the Reviewer for the thoughtful review and the constructive comments, which are greatly appreciated. The suggestions provided have meaningfully contributed to improving the clarity and structure of the manuscript.

- 1. Clarifying the Contribution of the Work: In line with the Reviewer's comment, and consistent with feedback from Referee #2, the manuscript has been revised to more clearly differentiate this study from the earlier work (Roded et al., 2024). Specifically, most of Section 2 ("Settings and Model Equations") has been relocated to Appendix A. The main Section 2 has been restructured to focus directly on the derivation and details of the equilibrium-based approach developed here. Section 2.2 now explains more clearly how the simplifying equilibrium assumption is implemented and how it differs from the kinetic framework in the previous paper.
- 2. Model Limitations Near Injection Well: I thank the Reviewer for highlighting the important issue of model limitations near the injection well, where the local

equilibrium assumption may break down. The manuscript was revised accordingly and discusses the origin and limitation of this discrepancy, referred to as the "inlet advective discrepancy" (lines 304-306). Under high Damköhler number conditions and quasi-equilibrium assumptions, deviations between the equilibrium and kinetic solutions are generally confined to a narrow zone near the injection point (see Figs. 2c and 3c in the newly added results for the planar case).

However, at very early times or under conditions farther from equilibrium (i.e., lower Damköhler numbers), the system is more likely to transition into a regime where the assumptions of the analytical equilibrium model no longer hold—particularly near the inlet. This breakdown is illustrated in Fig. 3a and is also captured by the applicability criterion derived in Section 3.3 (Eqs. 17 and 18). This consideration is particularly important in practical geothermal and hydrological contexts, where the relevant time frame may be limited to several years. The revised manuscript explicitly discusses these limitations and clarifies the domain of applicability of the equilibrium model (lines 300-314 and Section 3.3).

**Author Response to Reviewer #2's Comments**

This Author Response includes detailed responses to the Reviewer's comments, quoted verbatim in **bold**, along with the corresponding revisions made to the manuscript.

Dr. Roded presents an analytical solution-based analysis of reactive, thermal (horizontal, radial and planar) flow resulting from injection into a thin aquifer confined between impermeable rock. This is a follow-up to a previous study (Roded 2024b) and declared to analyze the effect of simplifying the full kinetic treatment with an equilibrium reaction assumption in cases that can be considered transport- rather than reaction rate-controlled.

While the overall results seem to be potentially useful, I have a number of critical remarks on this contribution and I strongly recommend applying major revisions to make this paper more approachable and useful for the community:

I am grateful to the Reviewer for his careful review and for the important and constructive comments. The detailed conceptual and technical feedback has been incorporated into the revised manuscript, and I believe it significantly improves the paper and enhances its clarity.

**Reviewer's Comment (1):**

Simple to correct, but utmost important: the units of the injection rate, Q, immediately appeared wrong. I quickly communicated with the author about this to simplify the review process and, indeed, instead of 500 m3 per second, they were taken as 500 m3 per day in the reported calculations (i.e., wrong by ca. 4 orders of magnitude). This needs to be corrected in Table 1 (easy) and I strongly recommend to also post a corrigendum to the previous paper where the same mistake occurs.

I thank the Reviewer for catching this important typographic error in the table. The values presented elsewhere in the text or figures, both in the previous work and in the current manuscript, have been verified to correctly indicate the intended value.

**Summary of revisions:**

• Table 1 has been corrected in the revised manuscript, and the manuscript was carefully revised to avoid additional typos.

• A corrigendum will be submitted to HESS for the previous publication.

**Reviewer's Comment (2):**

Not so simple and very important: Over large parts of the manuscript the difference to Roded (2024b) is not apparent and the texts are very similar and Figures identical or very similar as well. A revision should make much clearer how the simplifying equilibrium assumption is actually implemented and modifies the previous work. Namely, the derivation of the analytical solutions is essentially identical up to ca. equation 15 and then, without explanation, equations 16 and 17 appear from the same steps of derivation as in the previous paper but are supposed to represent (and look) something different. Here, an explanation of what is the key difference compared to previous work would help a lot (it probably requires very little but I can't really judge based on the information given). It should not be the job of a reader to spend an afternoon or two to re-assess the math and try to find out what is different. Even more importantly is the question to what degree the results are a fortuitous result of the reaction chosen.

I agree with the Reviewer that both the distinction from the prior work and the implementation of the equilibrium assumption needed to be made clearer. To better highlight the contribution of the current study and reduce redundancy, most of Section 2 ("Settings and Model Equations") was relocated to Appendix A. The main Section 2 was then revised to focus more directly on elements specific to this study. This restructuring reduces overlap in the main text while keeping the full model description readily accessible to readers.

Specifically, Section 2.2 ("The Equilibrium-Based Approach") in the revised manuscript now outlines the derivation of the thermo-hydro-chemical (THC) equilibrium model in greater detail. These revisions clarify how the simplifying equilibrium assumption is implemented and how it differs from the framework established in the previous work (specifically, see lines 167–169). Furthermore, it clarifies that the results are not an outcome of the specific reaction chosen, but depend on the Damköhler number (please see also the response to comment 3). Additionally, Fig. 1 was modified to better illustrate the Reactive Lauwerier Problem (RLP) under the equilibrium assumption.

Regarding the similarity to the previous study, I note that—had it not been for the time gap—this work and Roded et al. (2024) could have been published together as companion

papers. Since this was not possible, and to ensure the current manuscript stands on its own, some repetition of model settings and equations was necessary (though direct text duplication was avoided). Additionally, Section 3.2 ("Comparison to the Reference Solution") includes background information to facilitate comparison between the newly developed solutions and those of the previous work (the "reference solutions").

**Summary of revisions:**

- Large portions of Section 2 were relocated to Appendix A, while the main body of Section 2 was revised to focus on the key equations and elements directly relevant to this study. Particularly, it outlines the derivation of the THC equilibrium model (Eq. 5) and how it differs from the framework established in the previous work (see lines 167–169).
- The analytical derivations were revised for greater clarity (Section 3.1).
- Fig. 1 was modified to better illustrate the Reactive Lauwerier Problem (RLP) under the equilibrium assumption.

**Reviewer's Comment (3):**

Following up on that: I sometimes had a hard time understanding in what sense now the "equilibrium" assumption is to be understood. Why is a solution for the reaction rate needed (line 259) and its pre-requisites such as "solute disequilibrium" (eq. 7)? What is the meaning of the latter (also plotted in Fig. 2(b)) if equilibrium is supposed to be applicable? Should the arguments in Section 3.3 (but see criticism of that section in Point 5 below) come further upfront to make this clearer?

I thank the Reviewer for highlighting that these points were made not sufficiently clear in the manuscript. I believe that the revision of Section 2—and specifically Section 2.2 ("The Equilibrium-Based Approach")—now better explains these concepts. Specifically, the manuscript was revised to clarify the following key points:

- Under the equilibrium assumption, the solution is derived for the reaction rate, Ω, rather than for solute concentration (or disequilibrium, Λ). As such the equilibrium model eliminates the need for detailed kinetic formulations (see Eq. 5 with its associated text and also lines 131–135 and 125–126).
- In the derivation of the equilibrium model,  $\Lambda \approx 0$  is assumed (cf. Eqs. 4 and 5). This assumption is applicable for high Damköhler conditions and fast kinetics, where

quasi-equilibrium prevails and  $\Lambda \ll \Delta c_s$ . Here,  $\Delta c_s$  represents the absolute solubility change in the system ( $\Delta c_s = |c_s(T_{in}) - c_s(T_0)|$ , i.e., between  $c_s$  at the injection temperature and at ambient conditions).

Hence, a small  $\Lambda$  confirms the validity of this approximation and solute disequilibrium is used to quantify how closely equilibrium is maintained across space and time (see Figs. 2b, 3b, and 4a). Namely, even within an equilibrium-assumption framework, evaluating the disequilibrium,  $\Lambda(\xi, t)$ , is valuable for assessing model applicability. The disequilibrium-based solution (Eq. B3 in Appendix B) is also used in the derivation of the applicability criterion for the equilibrium approximation, as detailed in Section 3.3. These clarifications were added in lines 294–300 and 783–786.

Lastly, following these revisions intended to improve clarity, I believe it remains preferable to retain the current order of presentation—first presenting the results in Figs. 2-4, and then presenting the applicability criterion in Section 3.3—as this sequence offers a clear and logical progression for the reader.

**Reviewer's Comment (4):**

The manuscript would very much benefit if the "planar" flow results would also be visualized, as the resulting fronts look different from the radial case, i.e., the aquifer can be heated up to the injection temperature for significant distances, displaying a true "front" rather than an outward migrating diffusive-looking profile. Then, for both cases, it'd be interesting if a minimal phenomenological description of the underlying physics would be provided—why the progressing fronts have the shape that they have. I was a bit surprised that in the radial case it takes very long times to heat even the nearest region around the injection well to injection temperatures. Is that because the radial flow velocities slow down rapidly with radial distance, enhancing conductive heat loss relative to the planar case?

Following the Reviewer's comments, results of the planar case are now included in the revised manuscript. While the primary focus has been on the radial case, due to its greater relevance to geothermal systems and natural settings, I agree that the planar configuration is also of interest.

The Reviewer is correct in noting that, in the planar case, the aquifer can be heated to significantly greater distances. Regarding the difference in thermal front characteristics,

velocity decay in radial geometry enhances heat loss by conduction relative to the planar case. Specifically, in radial flow, the fluid velocity decreases with distance from the injection point, leading to longer residence times and greater heat transfer into the surrounding rock.

Additionally, in the radial case, the heat source (e.g., an injection well) acts as a source from which hot fluid spreads outward radially. In contrast, the planar configuration can be conceptualized as injection from a distributed source (e.g., a row of wells) generating a uniform planar front. More precisely, under the perfect thermal mixing assumption, the radial case is treated mathematically as a point source, while the planar case is treated as a line source. Hence, in the radial case, heat conduction is multidirectional, whereas in the planar case, heat is conducted only upward and downward. Of course, in both cases, the thermal front advancment is also influenced by flow rates and the volume of injected hot fluid.

In the revised manuscript, results for the planar case are now included in Fig. 3, similar to those presented in Fig. 2 for the radial case and above discussion has been added in lines 327–340. As the Reviewer expects, zooming in on the results in Fig. 3 at distances of around 2 km from the inlet shows that the aquifer can be heated to nearly the injection temperature over substantial distances. For the radial case, by contrast, effective heating near the injection well and later quick decay leads to a sigmoidal (or diffusive front-like) profile rather than a more simple decaying profile as seen in the planar case.

Additionally, given its importance for the investigation of coalesced front development, this topic is examined in detail in Section 3.4 ("Development of Coalesced Fronts"). Equations 22 and Fig. 5 present the evolution of the thermal front's outer edge,  $\xi_F$ , and illustrate the differences between the two cases. Here,  $\xi_F = r_F$  or  $x_F$  in the radial and planar cases, respectively, is defined as the downstream location where the temperature slightly deviates from ambient, via  $\varepsilon = (T(\xi_F) - T_0)/\Delta T$ , with  $\varepsilon \ll 1$  and  $\Delta T = T_{\rm in} - T_0$ .

The results in Section 3.4 show that the advancement rate of  $\xi_F$  decays much faster in the radial case, with  $\partial \xi_F/\partial t$  scaling as  $t^{-3/4}$ , whereas in the planar case it follows  $t^{-1/2}$ . This analytical result highlights the fundamental physical differences in front propagation between the two geometries.

Lastly, the results of Fig. 4 (previously Fig. 3), which illustrate the system's state evolution over time and the applicability of the equilibrium solution, are now also provided for the planar case in the Supplementary Material.

**Summary of revisions:**

- Results for the planar case, similar to those in Fig. 2, are now included in Fig. 3, with the relevant discussion in lines 327–340.
- Results corresponding to Fig. 4 (previously Fig. 3) are now included for the planar case in the Supplementary Material.
- The analytical discussion in Section 3.4 comparing the two geometries was revised to present and explain these differences more clearly (see lines 464–470).

**Reviewer's Comment (5):**

I had a hard time understanding the logic in sections 3.3 and 3.4. First, in my understanding, the main argument on which the analysis rests is not provided but referenced to an appendix of the previous publication (line 410). Why not simply and up-front repeat the key point from there that the thermal font advancement and the reactive front coalescence is simply due to the timescale involved and the nature of the reaction?

Then, there is often unclear writing and wording. For example, what is, in simple words, meant with "thermal front end location"? To me, it sounds like the thermal front comes to a final halt. But then it appears that this is rather meant as the furthest distance of thermal perturbation due to injection at a given time (the outer end of the non-sharp front)? But then, the definition in line 404 and the following manipulations don't seem convincing. First, there is not one position where the criterion is fulfilled but for a given  $\epsilon$  the condition is fulfilled for all r greater than a certain value (which I presume is this rF). Second, the choice of  $\epsilon$  would be arbitrary and, given the shape of the curve, potentially have a significant impact on the location of rF. I strongly suggest re-writing this whole part in much simpler and more approachable form. Please try to put yourself into the position of the reader who has not done your work and wants to understand what you did.

Another example for strange wording is the "elongation" in section 3.4. Fronts advance and broaden but don't "elongate". So, it seems you mean advancement, which would be a more commonly used term? And certainly not broadening, right? Although in the radial case it is a mixture of both ... Also, as it is written (e.g., lines 443, 447), at least for me, is comes across as if xF and xF refer to the

thermal and reactive fronts, respectively, while later it appears that these refer to the fronts in the planar and radial case. Please remove these ambiguities. There are plenty more such examples in the whole text, so a general overhaul towards clarity is highly desirable.

I thank the Reviewer for raising these critical points. Sections 3.3 and 3.4 have been rewritten and carefully revised to improve clarity and make the content more accessible. Please note that the subsectioning of Results Section 3 was revised as follows:

- Section 3.2 ("Comparison to Reference Solutions (High-*Da*)"): presents the comparison to the reference solutions at high *Da* conditions (previously Section 3.2.1).
- Section 3.3 presents comparison to the reference solutions at low *Da* conditions and the associated error along with the criterion for the applicability of the equilibrium solution. The full derivation of the criterion, which is more technical in nature, is now presented in Appendix C (the current Section 3.3 combines previous Sections 3.2.2 and 3.3).
- Section 3.4 as previously, analyze the development of the coalecensed fronts.

Regarding the main argument for the analysis of the coalesced fronts and the derivation of the criterion (Eqs. 17 and 18), the first paragraph of Section 3.3 was revised to present the central reasoning more clearly and explicitly (lines 437–441 and also in lines 763–767 of Appendix C). Specifically, the revised manuscript emphasizes that the coalescence of the reactive and thermal fronts is governed by the relevant timescales and high Damköhler number conditions. Under these conditions, any disequilibrium from thermal changes dissipates quickly and does not extend appreciably downstream, resulting in effective coalescence. With respect to the reference at line 410 to the previous publication, it serves only to cite the specific equation used in the analysis.

Concerning the definition of  $\varepsilon$  in line 404 of the previous version (i.e.,  $\varepsilon = (T(\xi_F) - T_0)/\Delta T$ , with  $\varepsilon \ll 1$ ), the text has been carefully revised to explicitly state that  $\varepsilon$  is a prescribed small value (here,  $\varepsilon = 0.01$ ) used consistently throughout the analysis. Specifically, it is substituted into the Lauwerier solutions (Eqs. 6 and 13) as well as Eq. B3. For a given prescribed  $\varepsilon$  (and corresponding parameter "a"), the definitions of  $\xi_F$  in Eqs. 21, 22, and C3 become unique.

While the choice of  $\varepsilon$  (restricted to values with  $\varepsilon \ll 1$ ) is somewhat arbitrary, the resulting expressions (Eqs. 17 and 18) remain valid and unaffected. Furthermore, the results shown in Fig. 4a and 4b, which are based on Eqs. 22, are not expected to change appreciably depending on the exact choice of  $\varepsilon$  (i.e., variations in  $\varepsilon$  and, correspondingly, in the parameter "a" would not meaningfully affect the results in Fig. 4a and 4b). The associated text was revised throughout the manuscript to explain this more clearly (lines 445–447, 451, 776-777, and 782).

The manuscript was also revised throughout for clearer and more consistent terminology. In particular, the word "elongation" was replaced consistently with "advancement," which more accurately reflects the intended meaning. Additionally, the ambiguity surrounding the definition of the front outer-end position variables— $x_F$  and  $r_F$  in the planar and radial cases, respectively—has been addressed by clearly and consistently defining these terms upon their first mention (lines 444–446 and 774–777).

**Reviewer's Comment (6):**

Generally, figure captions are too long and often too unclear. Too long: they should not contain explanation that belongs in the text (e.g., Fig 4). Too unclear: e.g., again Fig. 4: why don't you clearly state that panel (a) is for the planar and (b) for the radial case etc.? Or even put a respective label into the respective panels. Even with relatively careful reading it took me way too long to catch what "the low-flow-rate limit assuming conduction only" (panel C) should display. Most other figure captions show similar problems.

Figure captions throughout the manuscript have been shortened and revised to improve clarity. In particular, the captions were edited to focus on providing only the essential technical details directly related to the figures, while removing explanatory content.

Specifically, Fig. 5 was carefully revised to make it clearer—especially regarding the low-flow-rate limit now shown in panels (c) and (d). This case refers to the low-flow-rate limit in radial geometry, where conduction dominates and effectively distributes heat. It is now illustrated using two different approaches: (i) the analytical conduction-only solution, representing the limit as  $Q \rightarrow 0$  (black lines), and (ii) numerical results for low flow rates (Q = 1 and  $5 \, \text{m}^3$ /day, red and orange dashed and continues lines). Numerical results were added specifically to make the central point clearer and more convincing. Following the Reviewer's advice, labels were added directly to the panels to enhance clarity.

Please note that the captions of Figs. 1 and 5 have been shortened but remain somewhat lengthy due to the relatively large number of technical aspects that need to be clearly described for the reader.

**Reviewer's Comment (7):**

Very important: what is the relevance of the example calculations for real problems? Injection durations of 200, 10'000 and 100'000 years are unheard of and are very likely to be never implemented (for industrial-scale CO2 storage currently 25 years are typical projections); when extrapolating the magnified part of Fig. 2(c) to shorter times, it seems that one might, for real problems, enter a region where the analytical equilibrium solutions fail to be good approximations. Please discuss this.

I thank the Reviewer for these important comments. The manuscript has been revised to emphasize that the modeled scenario is not limited to engineered injection settings but is also relevant to natural systems, where flow and reactions can occur over geological timescales. This point was originally noted in lines 136–138 of Section 2.1 and is now also explicitly stated in lines 255–257 in the revised manuscript.

Additionally, the manuscript has been updated to include simulation results for a 25-year duration in Figs. 2 and 3 (replacing the former 200-year curve). These new results demonstrate the applicability of the equilibrium model even at early times (25 years).

However, as the Reviewer correctly notes, at even earlier times the system is more likely to enter a regime in which the assumptions of the analytical equilibrium model break down—particularly near the injection point. This limitation is consistent with the applicability criterion derived and presented in Eq. 17. The manuscript has been updated to explicitly discuss these limitations of the equilibrium solutions in the context of applied geothermal and hydrological systems (lines 426–435).

**Reviewer's Comment (8):**

The "reactions" are simple solubility curves without any actual reaction (e.g., a pH and aqueous speciation in the calcite case is excluded). It'd be desirable to see some reflection how applicable the solutions might be in more complex cases with multiple species and temperature dependencies of equilibrium constants that are less favorable for "coalescence" than in the calcite case. Nature has examples with multiple reactions fronts progressing at different velocities

(contact metamorphic aureoles, skarns in marble layers cross-cut by hydrothermal veins etc.), so these obviously didn't "coalesce" with the thermal front. To me, this makes the stated applicability to hydrothermal ore deposits questionable (where the author explicitly states: "A particularly intriguing phenomenon, often primarily controlled by the dependence of solubility on temperature, is the zoning of metals and minerals, which is commonly observed at various field scales. In these cases, regular belts of different precipitants form progressively as the distance from the hydrothermal fluid source increases." - so, how does this match the result of the present study with a single such precipitation front aligning with the thermal front being a key outcome?). This is particularly the case for porphyry copper deposits where the author seems to be not on top of the discussion. Already in 1992, Hemley and Hunt (Econ. Geol., 87, 23-43) specifically eluded on the role of precipitation fronts and their dependence on transport, heat transfer rates etc. Also, the geometry and hydrology of porphyry systems (e.g., Weis et al., Science, 2012) does not compare well with the problem studied here. Interestingly, in porphyry copper deposits, there are transport-limited reaction fronts, seen as alteration halo as these have been studied well (e.g., Cathles and Shannon, EPSL, 2007). For such reasons, I strongly suggest leaving out much of the rather speculative discussion related to natural examples in its current form. Rather provide some that clearly relate to the problem studied in the sense of Fig. 1 or not mention them.

Due to the simplifications required to derive closed-form expressions, analytical solutions typically apply directly only to a limited set of real-world cases—sometimes even to none. However, they remain valuable for developing fundamental conceptual understanding, identifying key functional relationships, and serving as important benchmark cases for validating numerical models that can address more complex scenarios.

I am grateful to the Reviewer for noting in detail, however, that the scenario considered in this study is too simplified to support extrapolated conclusions about ore deposit formation. Consequently, the discussion was revised to focus only on cases that more closely match the settings considered in this study, excluding the topic of ore deposit formation (Sections 4.1 and 4.2).

Regarding the applicability of the presented solutions to more complex cases—such as those involving multiple species and temperature-dependent equilibrium constants that do not favor coalescence—the current approach is limited. While Roded et al. (2024b)

addressed far-from-equilibrium conditions, the extension to more complex kinetic systems remains an open challenge. The possibility of extending these solutions in several directions should be investigated. In particular, this includes incorporating multiple species with complex kinetics and accounting for additional or more intricate couplings between variables and parameters.

In such cases, semi-analytical approaches could prove especially useful. In particular, given the quasi-static assumption of reactive flow, the governing equations for reaction rate (Eqs. 10 and 15) or solute disequilibrium (Eqs. B1 and B3) could be implemented in a semi-analytical, coupled, and iterative framework. Such an approach may allow the inclusion of multiple chemical species and more intricate couplings. However, its validity remains subject to further evaluation and future investigation.

Furthermore, the approach developed here and in Roded et al. (2024b) can be adapted to extend additional thermal solutions to significant thermally driven reactive transport scenarios. Notably, this may be especially practical under the equilibrium assumption, where thermally driven reactions depend primarily on thermal gradients.

This expanded discussion of these limitations and the theoretical modeling outlook is now included in the revised manuscript (Section 4.3, "Theoretical Modeling Outlook").

**Summary of revisions:**

- The discussion was revised to focus only on cases that more closely match the settings considered in this study, excluding the topic of ore deposit formation (Sections 4.1 and 4.2).
- A discussion of possible pathways to extend the solutions to more complex systems and future research directions has been added (Section 4.3, "Theoretical Modeling Outlook").

**Final Reviewer's Comment:**

In spite of this criticism, I found the work quite inspiring. The more it seems important to make it more approachable and clear such that people can benefit from it.

I once again thank the Reviewer for the careful review, very insightful and important comments, and positive attitude. I hope that the carefully revised manuscript is improved,

clearer, and more accessible to a broader readership, and I welcome any further adjustments.